# The association between the error-related negativity and self-control is moderated by impulsivity and compulsivity
Rebecca Overmeyer [1] ✉, Anja Kräplin [2], Thomas Goschke[1,3] & Tanja Endrass [1,3]

Impaired self-control has been linked to deficits in performance monitoring and is associated with impulsive and compulsive behaviors. Although altered error-related negativity (ERN) amplitudes have been observed in disorders characterized by these traits, it remains unclear how such alterations relate to the translation of monitoring signals into behavior. In a sample of 221 participants, we combined electroencephalography with ecological momentary assessment to examine how self-reported impulsivity and compulsivity relate to daily self-control and moderate the association between ERN amplitude and daily self-control. High compulsivity was associated with increased desire enactment and more frequent self-control failures. Critically, ERN amplitude was associated with better self-control only at low levels of both impulsivity and compulsivity. These findings indicate that the relationship between performance monitoring and daily self-control varies as a function of trait-level impulsivity and compulsivity. While the present findings are correlational in nature, they suggest that considering trait interactions may inform cognitive and neural models of self-control and our understanding of how performance monitoring translates into adaptive behavior.

Self-control is essential for achieving goals when competing desires arise. In cognitive-behavioral therapy, interventions often aim to strengthen self-control to improve goal-directed behavior, as impaired goal-directed control is a key feature of many mental disorders[1,2]. Successful self-control emerges from complex interactions between different neural processes representing internal action values, continuous behavior monitoring, and the recruitment of cognitive control [e.g., ref. 2]. Impaired self-control may result from dysfunctions at different levels of processing. Understanding these impairments is crucial for a nuanced view of failures in self-control and impairments in goal-directed behavior.

Self-control refers to the ability to regulate behavior, thoughts, and emotions in accordance with a specific goal or personal standard by changing or overriding competing response tendencies, desires, or temptations[3,4]. Self-control includes both inhibitory and initiatory components and is crucial for adaptive behavior and successful goal pursuit, especially in contexts influenced by affective states[5–8]. A broader concept closely related to self-control is self-regulation, which includes goal setting, monitoring for the need of control, and implementing control processes according to set goals[9,10]. State self-control is often parsed and assessed using ecological momentary assessment of desires, associated goal conflict, and behavioral outcomes[4,11,12], showing that higher desire strength was predictive of

behavior enactment, whereas higher conflict strength was predictive of the behavior not being enacted.

Neural networks underpinning self-control include the valuation network, which integrates action values and guides value-based decision-making[13,14], the performance monitoring network, which detects conflicts between goals and desires or selected actions[15,16], and the cognitive control network, which maintains goals and anticipates outcomes[17]. When a conflict is detected by the performance monitoring network, control is recruited to support goal-directed behavior. Effective self-control thus depends on performance monitoring, including error processing, which signals deviations of response outcomes from intended outcomes and is thought to alert the cognitive control system to the need for adaptive regulation[15]. Performance monitoring is theorized to guide action selection by signaling the need for cognitive control.

Deficient performance monitoring may be related to impaired recruitment of cognitive control[18], lower top-down regulation, and lower impact of long-term goals on decision-making, which may be associated with increased impulsive or maladaptive behavior[2,13]. Previous research has provided some evidence for this hypothesized link between performance monitoring and self-control: higher levels of self-control failure, as measured by ecological momentary assessment, were associated with lower

[1]Faculty of Psychology, Dresden University of Technology, Dresden, Germany. [2]Department of Psychiatry and Psychotherapy, Dresden University of Technology, Dresden, Germany. [3]Neuroimaging Centre, Dresden University of Technology, Dresden, Germany. ✉e-mail: rebecca.overmeyer@tu-dresden.de

error-related activation of the performance monitoring network[19] and reduced error-related negativity (ERN) amplitude[20].

Impaired self-control and heightened influence of desire may reflect deficits in signaling the need or the implementation of cognitive control, affecting both healthy and clinical populations[2,21,22]. Failure of self-control can manifest in impulsive and compulsive behaviors[23,24]. While overlapping, impulsivity and compulsivity are distinct multidimensional traits[25]. Impulsivity involves rash, poorly planned actions and increased shifting, while compulsivity reflects rigid, perseverative behaviors emphasizing stability[26,27]. When excessive, these behaviors can become inflexible and disconnected from their intended goals, and these deficits in goal-directed control may play a central role in the development of mental disorders[1], such as obsessive-compulsive disorder (OCD) and substance use disorder (SUD). For example, individuals may persist in substance use despite diminishing pleasure and in disregard of long-term consequences and goals, as craving continues and intensifies even when hedonic effects weaken[28].

Consistent with this, individuals with mental disorders characterized by deficient goal-directed control, such as those showing high compulsivity and impulsivity, exhibit altered neural correlates of performance monitoring, particularly the ERN[28–31]. This aligns with the role of performance monitoring in the recruitment of control[32], as larger ERN amplitudes are associated with fewer self-control failures[20]. Higher ERN amplitudes may reflect a heightened response to errors, signaling a greater demand for cognitive control, which may be associated with more effective self-control[32,33]; higher ERN amplitudes have been shown to predict better stress regulation[34–36] and self-regulation skills[37]. However, motivational factors also modulate this process[38,39]: According to the Expected Value of Control theory, the mobilization of cognitive control in response to errors depends on the subjective evaluation of whether exerting control is worth exerting[40].

Although higher ERN amplitudes are generally linked to better self-control, findings in individuals with OCD challenge this association. Individuals with OCD display elevated ERN amplitudes[41], but also exhibit impaired self-control in daily life. Additionally, these larger ERNs are often decoupled from task performance and within-task behavioral adaptations like post-error slowing or sensitivity to motivational contexts[29,33]. This dissociation suggests that while performance monitoring is heightened, its translation into behavioral adaptation is disrupted, leading to failed or maladaptive control implementation[33,42]. For example, individuals with OCD may exert self-control in maladaptive ways, such as compulsive checking of the stove to prevent a house fire[22,43]. In fact, this behavior reflects a heightened need for control with a reduced sense of actual control, contributing to impaired self-control, e.g., in the form of increased checking behavior[44,45]. A strong desire for control may also amplify conflicts between goals and perceived performance potential, potentially contributing to heightened monitoring activity[46]. This is particularly relevant, as the connection between ERN and adaptive behavior is likely moderated by trait dimensions rooted in affective-motivational dysregulation, as pointed out by Bartholow[47].

In contrast, disorders characterized by impulsivity, such as SUD or gambling disorder, are typically marked by underregulation of control processes[48]. Individuals with these conditions tend to experience stronger or more frequent temptations[49,50]. The lower ERN amplitudes observed in such individuals may reflect reduced internal conflict about acting on such desires[30,31], possibly because they assign greater subjective value to short-term response options[51]. Extending this, larger ERN amplitude has been shown to predict less risky alcohol use in adolescents[52] as well as successful abstinence after treatment in cocaine use disorder[53]. Furthermore, high impulsivity has been associated with diminished modulation of ERN amplitude by motivational context, particularly showing attenuated ERN in punishment-motivated trials compared to individuals with low impulsivity[54]. These findings underscore that self-control is not simply a matter of more or less control, but requires a nuanced understanding of how much self-control is adaptive, and when behavior becomes excessive, misguided, or costly[2].

To address these complexities and clarify the association between error monitoring and self-control in mental disorders, this study investigates how impulsivity, compulsivity, and related psychopathological symptoms affect daily self-control and modulate the association between ERN and self-control. Using a mixed-method approach, we investigate whether the association between error monitoring (ERN amplitude) and daily self-control is influenced by self-reported psychopathological symptoms. Self-control is measured using ecological momentary assessment (EMA) over a 7-day period [similar to ref. 55], tracking desire occurrence, goal-desire conflict, and desire enactment, and thereby providing a process-oriented account of interventive self-control that allows self-control failures and successes to be examined dynamically in daily life. Psychopathological symptoms are assessed with validated self-report instruments. Performance monitoring, specifically the ERN, is measured using a modified arrow Flanker task[56,57] with gain and loss contexts to assess motivational effects. The study focuses on how impairments in goal-directed control relate to different psychopathological symptom dimensions and their interaction.

We preregistered the following hypotheses (https://doi.org/10.17605/OSF.IO/Y8R27; Table 1 provides the preregistered hypotheses alongside their corresponding formulation in the manuscript): (1) Higher levels of compulsivity and impulsivity will be associated with higher desire enactment and lower self-control. Compulsivity and impulsivity will moderate the association between desire strength and conflict strength with enactment: While high compulsivity will be associated with a weaker influence of desire strength and/or a stronger influence of conflict strength, high impulsivity will be associated with a stronger influence of desire strength and/or a weaker (negative) influence of conflict strength. (2) Compulsivity and impulsivity will influence the reporting of desires and the experience of conflict strength: High compulsivity will be associated with reporting fewer desires, but more conflict about these desires, while high impulsivity will be associated with experiencing more desires, but less conflict about these desires. (3) Compulsivity and impulsivity will moderate the association between the ERN and self-control failures: High compulsivity will be associated with a diminished association between the ERN and enactment, reflecting unsuccessful application of self-control. High impulsivity will be associated with a negative ERN-desire enactment relationship in gain contexts (i.e., greater ERN predicts less enactment), but a reduced association in loss contexts. (4) In an exploratory analysis, we also examined whether participant clusters based on psychopathological symptom profiles (beyond impulsivity and compulsivity) differ in self-control.

## Methods

Performance monitoring was assessed using an arrow flanker task as part of an EEG session in the lab; self-control was assessed using EMA during a 7-day period. Questionnaire data were obtained at the EEG session and at an earlier lab appointment. See Fig. 1 for an overview. Participants also completed other EEG as well as behavioral tasks that are not reported here.

Hypotheses and analysis plans (confirmatory as well as exploratory) were preregistered on the Open Science Framework (OSF; https://doi.org/10.17605/OSF.IO/Y8R27), where all materials are also available, after data acquisition, but before analysis. For clarity and improved readability, we paraphrased the hypotheses in the main text and reorganized them by dependent variable/analysis rather than by trait (impulsivity/compulsivity). Table 1 provides a comparison between the original preregistered hypotheses and respective formulations used above. All procedures complied with the ethical guidelines of the Declaration of Helsinki and were approved by the Ethics Committee at the University Hospital, at the Dresden University of Technology (EK 372092017). All participants provided written informed consent. Participants were compensated with up to €100 depending on task performance and compliance. A compliance-based bonus (€10) was provided for conscientious completion of the smartphone-based assessment, and a performance-based bonus of up to €5 could be earned in the Flanker task.

### Power analysis
The power analysis was designed to ensure sensitivity for detecting small effects involving three predictors central to the study's hypotheses:

**Table 1 | Transparency notes for hypotheses (https://doi.org/10.17605/OSF.IO/Y8R27)**

| Preregistered hypotheses (verbatim) | Manuscript formulation (paraphrased) |
|---|---|
| Impulsivity:<br>1. High impulsivity leads to more self-control failures (enacted conflict-laden desires).<br>Compulsivity<br>1. High compulsivity leads to more self-control failures (enacted conflict-laden desires). | 1) Higher levels of compulsivity and impulsivity will be associated with higher desire enactment and lower self-control. |
| Impulsivity:<br>2. High impulsivity leads to a stronger (positive) influence of desire strength on the enactment of conflict-laden desires and/or a weaker (negative) association between conflict strength and the enactment of conflict-laden desires.<br>Compulsivity<br>2. High compulsivity leads to a weaker (positive) influence of desire strength on the enactment of conflict-laden desires and/or a stronger (negative) association between conflict strength and the enactment of conflict-laden desires. | Compulsivity and impulsivity will moderate the association between desire strength and conflict strength with enactment: While high compulsivity will be associated with a weaker influence of desire strength and/or a stronger influence of conflict strength, high impulsivity will be associated with a stronger influence of desire strength and/or a weaker (negative) influence of conflict strength. |
| Impulsivity:<br>3. High impulsivity is associated with experiencing more desires, but less conflict about these desires.<br>Compulsivity<br>3. High compulsivity is associated with experiencing fewer desires, but more conflict about these desires. | 2) Compulsivity and impulsivity will influence the reporting of desires and the experience of conflict strength: High compulsivity will be associated with reporting fewer desires, but more conflict about these desires, while high impulsivity will be associated with experiencing more desires, but less conflict about these desires. |
| Impulsivity:<br>4. High impulsivity is associated with a negative association between self-control failures and the ERN in a context of potential gain. For potential loss, high impulsivity may lead to a diminished association between ERN and self-control failures.<br>Compulsivity<br>4. High compulsivity is associated with a negative but diminished association between error-related brain activity and self-control failures, reflecting unsuccessful application of self-control. | 3) Compulsivity and impulsivity will moderate the association between the ERN and self-control failures: High compulsivity will be associated with a diminished association between the ERN and enactment, reflecting unsuccessful application of self-control. High impulsivity will be associated with a negative ERN-desire enactment relationship in gain contexts (i.e., greater ERN predicts less enactment), but a reduced association in loss contexts. |

impulsivity, compulsivity, and the ERN. Their joint influence on self-control was captured by their three-way interaction (ERN × impulsivity × compulsivity) in the primary models. The effect size was set at $f^2 = 0.05$, slightly smaller than reported in previous research on the connection between error-related brain activity and self-control[19] ($\beta = 0.25$, corresponding to $f^2 \approx 0.07$), to ensure adequate power. Detecting this effect size with 80% power and an alpha level of 0.05 (two-tailed) required a sample size of 223 participants. To account for potential dropouts due to attrition or data quality, we recruited 253 participants, exceeding this requirement.

## Sample

We recruited 253 participants from the general Dresden area to cover a range of self-reported compulsivity and impulsivity as assessed by the OCI-R[58,59] and the BIS-11[60]. All participants were fluent in German, reported no history of head trauma or neurological disease, had normal or corrected-to-normal vision, and were abstinent from all psychotropic substances, except for nicotine and alcohol, within the previous 3 months, in addition to not using illicit substances more than twice a year or cannabis more than twice a month. They were not included in the study if they reported a history of emotionally unstable personality disorder, bipolar disorder, severe alcohol use disorder, or psychotic episodes; if they currently met the criteria for an eating disorder or a severe episode of major depression. We excluded 32 participants after data collection for the following reasons: poor EEG data quality ($n = 1$), discontinuation of assessment ($n = 1$), not following task instructions as indicated by excessive random button presses during the task ($n = 7$), fewer than 16 errors ($n = 11$) per motivational context[61], and an error rate exceeding 40% ($n = 12$). The final number of participants was 221 ($M = 25.16$ years, $SD = 4.94$). Of these, 46.6% ($n = 103$) self-reported as female and 53.4% ($n = 118$) self-reported as male. A total of 93.7% ($n = 207$) had completed higher education degrees, and 11.3% ($n = 25$) reported a history of mental health problems. Participants self-reported ancestry as mainly European (91.9%, $n = 203$), mainly Asian or Middle Eastern (5.9%, $n = 13$), or unknown (2.3%, $n = 5$). Results on a subsample[20,62] and analyses regarding different research questions within this sample[63] have been published before.

## Flanker task

A modified version of the arrow-version of the Eriksen flanker task[56,57] with a potential gain and a loss avoidance incentive context[62] (see also Fig. 1), was employed to assess performance monitoring. Participants were instructed to respond as quickly and accurately as possible by pressing either a left or a right button, depending on the direction of a centrally presented target arrow. The target stimulus was displayed for 30 ms, with a 100 ms delay relative to the onset of the surrounding flanker arrows. In 50% of the trials, the target arrow pointed in the opposite direction (incongruent) to the flanker arrows, while in the other 50% of the trials, the target arrow pointed in the same direction (congruent). Each trial began with an incentive cue, signaling a potential gain or loss of 40 points. This cue was represented by a red (loss) or green (gain) frame surrounding a fixation cross and remained visible for 500 ms. The frame remained for the duration of the trial. After the participant's response, performance feedback was provided. Negative feedback was provided for incorrect responses and for the slowest 20% of correct responses, while positive feedback was provided for the remaining correct responses. In the loss context (50% of the trials), negative feedback resulted in a loss of 40 points, whereas positive feedback indicated no loss (0 points). In the gain context, positive feedback resulted in a gain of 40 points, whereas negative feedback indicated no reward (0 points). The response deadline was adaptively set based on individual performance and response time to achieve a rate of 20% negative feedback within each context. Performance feedback was presented for 800 ms, following a response interval of either 900 ms after target onset or 600 ms after the response. Participants accumulated points and could earn a bonus of up to 5 EUR based on their performance. The task consisted of 640 trials, each lasting between 2.53 and 2.75 s, in addition to a training block of 40 trials. The total task duration was approximately 25 min and was presented using Presentation 19.0 (Neurobehavioral Systems Inc., Berkeley, CA, USA). All materials and further details are publicly available alongside the preregistration (https://doi.org/10.17605/OSF.IO/Y8R27).

## Questionnaires

*Impulsivity* was assessed using the 11th version of the Barratt Impulsiveness Scale[60,64] (see Supplementary Note 1.2). *Compulsivity* was assessed using the

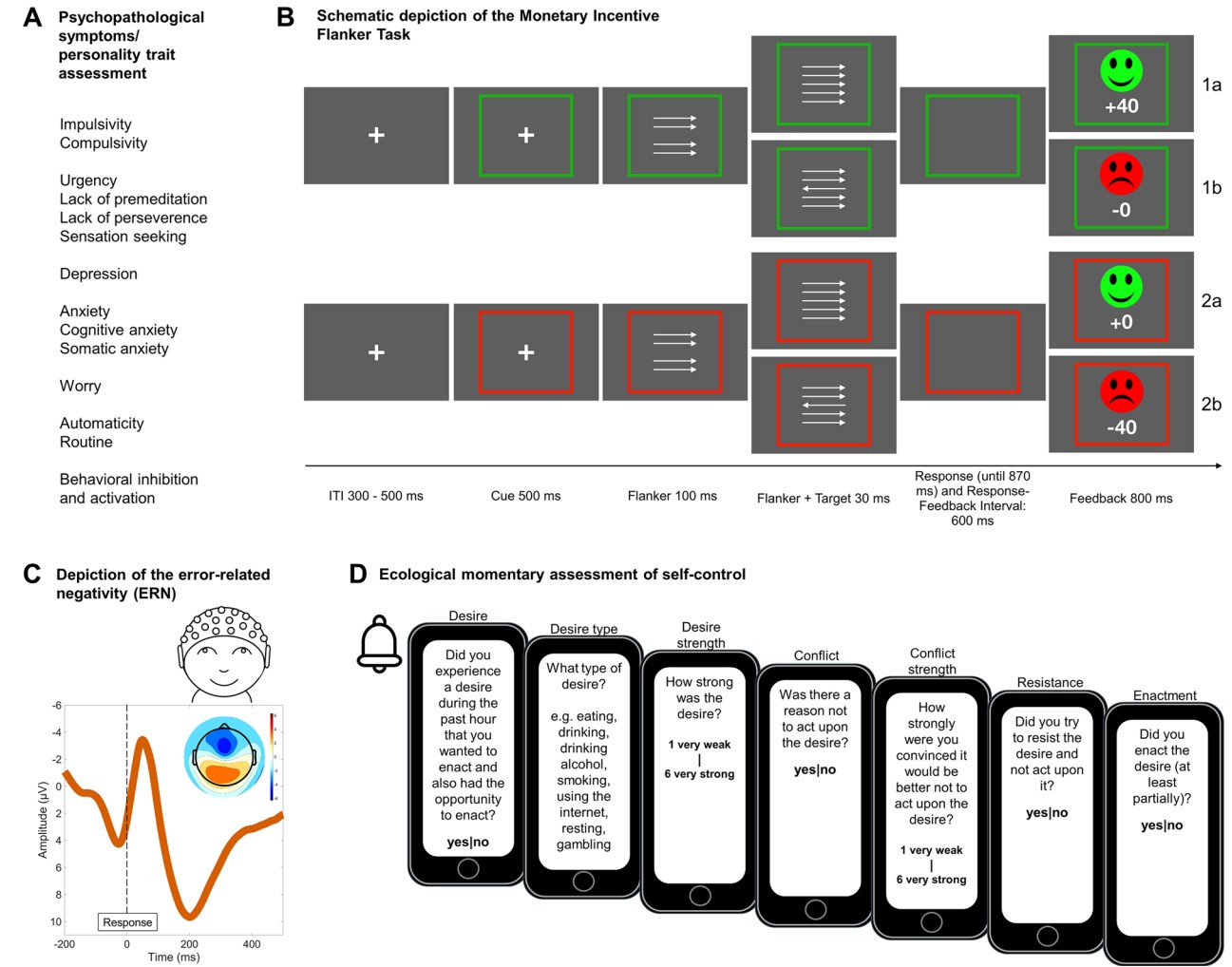

**Fig. 1 | Overview of utilized measures, including questionnaires, Flanker task, error-related negativity, and ecological momentary assessment.**
**A** Psychopathological symptoms and personality traits potentially related to goal-directed control assessed in the laboratory. Impulsivity: Barratt Impulsiveness Scale (BIS-11); compulsivity: Obsessive-Compulsive Inventory-Revised (OCI-R); urgency, lack of premeditation, lack of perseverance, sensation seeking: four sub-scales of the UPPS Impulsive Behavior Scale (UPPS); Depression: depression sub-scale of the Depression Anxiety Stress Scales; Anxiety: anxiety subscale of the Depression Anxiety Stress Scales (DASS-21); Cognitive and somatic anxiety: cognitive and somatic subscales of the State-Trait Inventory for Cognitive and Somatic Anxiety (STICSA); Worrying: Penn State Worry Questionnaire (PSWQ); Automaticity and routine: subscales of the Creature of Habit Scale assessing facets of habitual propensity (COHS); Behavioral inhibition and activation: BIS/BAS scales. See Supplementary Note 1.2 for details. **B** Task structure of the Monetary Incentive Flanker task (MIFLAT) used to assess the ERN as a marker of performance monitoring. Participants had to respond with the left or right button, in accordance with the direction of the middle arrow. If participants responded correctly in the gain context indicated by the green box, the 80% fastest responses were rewarded (1a), in the loss context indicated by the red box, the 80% fastest correct responses were not punished (2a). The 20% slowest and incorrect responses were either not rewarded (gain context, 1b) or punished (loss context, 2b). ERN: error-related negativity. Reproduced from Overmeyer et al.[20], under the terms of CC BY 4.0. **C** Depiction of the error-related negativity (ERN), averaged within the sample. Time course of mean response-locked EEG activity at electrodes Fz, FCz, and Cz for incongruent error trials for the analyzed sample ($n = 221$). The topographical map of response-locked EEG activity for incongruent error trials around the mean ERN latency at 54 ms ± 10 ms is depicted on the upper right. **D** Structure of the questionnaire used to assess self-control in daily life. The questionnaire was delivered 8 times per day for 7 consecutive days. We assessed the desire occurrence, their subjective intensity, if the desires were conflict-laden, how strong the conflict was, if participants tried to resist the desire, and if they enacted the desire. Four dichotomous variables (desire, conflict, resistance, and enactment), one categorical variable (desire type), and two continuous variables (desire and conflict strength) were obtained per questionnaire.

Obsessive-Compulsive Inventory-Revised[65] (see Supplementary Note 1.2). Additional impulsivity facets were assessed using the Urgency, Premeditation (lack of), Perseverance (lack of), Sensation Seeking, Impulsive Behavior Scale[66,67] (see Supplementary Note 1.2). Depression, anxiety, and worry were measured using the Depression Anxiety Stress Scales[68,69] (see Supplementary Note 1.2), the State-Trait Inventory for Cognitive and Somatic Anxiety[70,71] (see Supplementary Note 1.2), and the Penn State Worry Questionnaire[72,73] (see Supplementary Note 1.2). Habitual propensity was operationalized using the Creature of Habit Scale[74,75] (see Supplementary Note 1.2). Behavioral inhibition and activation were measured using the BIS/BAS scales[76,77] (see Supplementary Note 1.2).

**Ecological momentary assessment**
We employed ecological momentary assessment (EMA) to measure self-control in daily life. Consistent with recent distinctions between self-control capacity, urge intensity, and behavioral outcomes[11], the present study did not measure self-control capacity directly. Instead, we adopted a process-oriented approach. Participants were provided with identical smartphones (Nokia 5) with all functions disabled except for a customizable EMA application (movisensXS, version 1.3.3; movisens GmbH, Karlsruhe, Germany). After completing a tutorial to ensure understanding of the questionnaire, they received eight brief questionnaires per day over a 7-day period, allowing up to 56 completed questionnaires per participant,

**Table 2 | Transparency notes and deviations from the preregistration: (https://doi.org/10.17605/OSF.IO/Y8R27)**

| Preregistration | Manuscript |
|---|---|
| A highly complex multilevel model with five-way interactions (e.g., desire_strength × conflict_strength × impulsivity × compulsivity × ERN) was preregistered. | Model complexity was reduced to improve statistical power, enhance model convergence, avoid overfitting, and retain interpretability. |
| Analyses focusing on enactment were preregistered to be based either on all situations or only on conflict-laden situations. | Our hierarchical multilevel models included all reported desire episodes, not only those involving reported conflict, although we also preregistered hypotheses on self-control failures (i.e., enactment of conflict-laden desires). This preregistered and theoretically motivated approach enabled us to examine both the emergence of a goal-conflict and the failure to regulate conflicting desires[4]. Modeling conflict strength as a continuous Level-1 predictor captures how impulsivity and compulsivity shape different stages of self-control. Importantly, our preregistered analyses of conflict-laden desires yielded similar results, demonstrating robustness across operationalizations of self-control behavior. Moreover, because compulsivity—but not impulsivity—predicted conflict experiences, restricting analyses to conflict-laden episodes would have biased results toward high-compulsive individuals. Thus, our modeling approach reflects the full range of trait variability and the multi-stage nature of self-regulation in daily life. |
| Regression mixture models, including ERN to explore individual differences in ERN–self-control associations, were planned. | Regression mixture models were performed without ERN, focusing instead on clusters predicting self-control behavior, to improve statistical power. We then explored the moderating effect of cluster membership on the association between ERN and self-control failures. |

depending on response rates. Questionnaires were delivered at randomized times, at least 1 h apart, and participants could manually postpone alarms for up to 15 min. The 14-h delivery window began at 8, 9, or 10 a.m., based on individual waking hours. This procedure was similar to that used by Wolff et al.[55] and Hofmann et al.[4], where self-control is conceptualized as the exertion of control in the face of competing desires and long-term goals or a personal standard. Each questionnaire assessed whether a desire had been present within the preceding 60 min, the type of desire (see Supplementary Note 5.1 for a descriptive analysis), the strength of that desire, if there was a conflict in enacting the desire, the strength of the conflict, if participants tried to resist the desire, and whether the desire was ultimately enacted. For each questionnaire, we collected up to four dichotomous variables (desire occurrence, conflict occurrence, resistance occurrence, and enactment), one categorical variable (desire type), and two continuous variables (desire strength and conflict strength).

**Psychophysiological recording and data reduction**
EEG activity was recorded continuously from 61 sintered Ag/AgCl electrodes at equidistant locations in elastic EEG caps (EasyCap GmbH, Herrsching-Breitbrunn, Germany). Eye movement was recorded using two external electrodes placed below the right and left eyes, as well as electrodes at LO1 and LO2 mounted in the cap. Ground and reference electrodes were located next to Fz (at AFF1h and AFF2h, theta/phi spherical coordinates: −58/78 and 58/78). Data were recorded using high-pass and low-pass filters set at 0.1 to 250 Hz (Slope: 12 dB/Oct). Impedances were monitored during recording and kept below 10 kΩ. The continuous signal was digitized at a sampling rate of 500 Hz using two 32-channel BrainAmp amplifiers (Brain Products GmbH, Munich, Germany). All subsequent preprocessing was performed offline using EEGLAB[78] and MATLAB 2018b (The MathWorks Inc., 2018) and was preregistered before data analysis.

The data was high and low pass filtered using a Hamming windowed-sinc finite impulse response (FIR) filter with the lower edge of the pass band at 0.1 (cut-off frequency (−6 dB): 0.05 Hz) and 30 (cut-off frequency (−6 dB): 33.75 Hz) Hz[79]. The continuous data were then epoched from −500 to 2000 ms relative to target stimulus onset. Signal deviations of more than 5 SD of the mean probability distribution on any single channel or the whole montage were used as an artifact criterion to automatically reject epochs with artifacts. Remaining epochs were demeaned. Adaptive mixture independent component analysis (AMICA), implemented in EEGLAB 14.1.2[78], was used to identify independent components reflecting ocular or cardiovascular artifacts, which were manually identified, component loadings were then transferred to continuous data, and artifact components were removed (average number of components removed per participant = 4.74). The EEG data were then re-referenced to a common average reference. We chose the average reference scheme because we used a high-density montage and aimed to reduce bias from a single reference electrode. Subsequently, response-locked epochs from −500 to 1000 ms were created. The average EEG activity from 400 to 0 ms prior to the response was used as the baseline. We then calculated individual participant's ERP amplitudes per trial type. The ERN and CRN were calculated as the mean amplitude for ±10 ms around the individual negative peak in a cluster of frontocentral electrodes Fz, FCz, and Cz within 200 ms after the response.

**Data analysis**
All statistical analyses were performed using R[80]. Hierarchical linear models were fitted using the R package *lme4* version 1.1-29[81], and general linear models were fitted using the R package *stats*[80]. Model assumptions for hierarchical models were evaluated with the R package *DHARMa* version 0.4.6[82], which performs simulation-based residual diagnostics. Diagnostics indicated no evidence of overdispersion at the observation level (dispersion = 0.97, $p = 0.432$) and no substantial dispersion at the cluster level (dispersion = 0.74, $p = 0.056$), suggesting that binomial variance assumptions were adequately met. Visual inspection of random effects revealed no substantial deviations from normality. Multilevel models were estimated using maximum likelihood (ML) and the BOBYQA optimizer. Standardized parameters were obtained by fitting the model to a standardized version of the dataset. 95% Confidence Intervals (CIs) and $p$-values were computed using a Wald z-distribution approximation. Missing or skipped EMA prompts were handled via listwise deletion at the level of individual observations, corresponding to the default behavior of *glmer* in R[81]. Robustness checks (see Supplementary Note 3.2) were conducted for the multilevel models and indicated stable parameter estimates. For general linear models, assumptions of normality, homoscedasticity, and independence of residuals were generally met. Robust standard errors were applied when heteroscedasticity was detected, which was necessary only for the model testing the influence of the CRN on self-control (see Supplementary Note 4.1). Simple slopes were tested using the *emmeans* R package version 1.8.3[83]. All continuous regressors were scaled prior to analysis. ERN amplitudes were averaged across all errors in incongruent trials per participant and entered into the models as participant-level predictors. For analyses stratified by motivational context, ERN amplitudes were averaged separately for gain and loss contexts and entered accordingly. Data and code for the analyses are available on the OSF. For further transparency notes and deviations from the preregistration, see Table 2.

**The influence of impulsivity and compulsivity on self-control**. We used logistic multilevel regression analyses to examine how impulsivity and compulsivity impact self-control. We fitted hierarchical linear models, with situations (Level 1, experience sampling) nested within participants (Level 2). Level 1 predictors were desire and conflict strength[84], level 2 predictors were impulsivity and compulsivity; full model, in R's lme4[81] syntax: (1) *enactment ~ desire strength + conflict strength + impulsivity + compulsivity + desire strength:impulsivity:compulsivity + conflict strength:impulsivity:compulsivity (1 + desire strength + conflict strength | participant)*. We fitted two follow-up models to disentangle the interactions of impulsivity and compulsivity with desire and conflict strength: (2) model 1: *enactment ~ desire strength + conflict strength + impulsivity + compulsivity + desire strength:impulsivity + conflict strength:impulsivity (1 + desire strength + conflict strength | participant)*; (3) model 2: *enactment ~ desire strength + conflict strength + impulsivity + compulsivity + desire strength:compulsivity + conflict strength: compulsivity (1 + desire strength + conflict strength | participant)*.

**The influence of impulsivity and compulsivity on desire and conflict**. To analyze how impulsivity and compulsivity impact desire occurrence, conflict strength, and conflict reporting, we used generalized linear models (GLM): (4) *desire ~ impulsivity + compulsivity + impulsivity:compulsivity*; (5) *conflict strength ~ impulsivity + compulsivity + impulsivity:compulsivity*; (6) *conflict ~ impulsivity + compulsivity + impulsivity:compulsivity*.

**Moderation effects on the association between the ERN and self-control**. We examined how impulsivity and compulsivity influence the previously established association between the ERN and self-control in daily life using logistic multilevel regression analyses. We fitted hierarchical linear models, with situations (Level 1, experience sampling) nested within participants (Level 2). Level 1 predictors were desire and conflict strength[84], level 2 predictors were impulsivity and compulsivity, as well as the ERN: (7) base model: *enactment ~ desire strength + conflict strength + ERN + (1 + desire strength + conflict strength | participant)*; (8) full model: *enactment ~ desire strength + conflict strength + ERN + impulsivity:compulsivity:ERN + (1 + desire strength + conflict strength | participant)*. Thus, we examined the moderation effect while controlling for desire and conflict strength. This was examined in separate models for each motivational context.

**Establishing consistency with previous analyses**. For consistency with previous analyses[20], we additionally calculated GLMs to predict self-control failures operationalized as enactments of conflict-laden desires divided by the number of questionnaires participants had responded to (with predictors impulsivity, compulsivity and ERN, depending on motivational context: (9) *self-control failures ~ impulsivity + compulsivity + impulsivity:compulsivity* (full model); and (10) *self-control failures ~ ERN + impulsivity:compulsivity:ERN* (full model)).

**Profile regression to explore potential subgroups differing in self-control**. To explore if there are clusters defined by symptom dimensions or personality characteristic expressions relevant to adaptive behavior that differ in self-control, we used Bayesian profile regression, implemented as a Dirichlet process regression mixture model. Regression mixture models enable a search for evidence of heterogeneity in the effects of a predictor on an outcome: they explore the data for groups of participants who differ in these effects[85]. We used an analysis strategy similar to that of Dennison et al.[86].

We used the R package *cluster* version 2.1.6 and its function *pam*[87], and the R package *factoextra* version 1.0.7 and its function *fviz_nbclust* for the pam method[88], as an exploratory heuristic to inform the expected number of clusters, via partitioning around medoids. This is a more robust version of k-means[89]. After exploring the expected number of clusters, we performed

profile regression, which non-parametrically links the outcome to covariates through cluster membership[90]. This is implemented in the *PReMiuM* package version 3.2.13 in R[90], which implements profile regression as a Bayesian Dirichlet process regression mixture model. Individuals are probabilistically assigned to latent clusters that differ in both covariate distributions and outcome measures.

The outcome was operationalized as enactments of conflict-laden desires divided by the number of questionnaires participants had responded to; the explanatory variables considered in the clustering were: impulsivity and compulsivity, and additionally urgency, lack of premeditation, lack of perseverance, sensation seeking, depression, anxiety (assessed using three different scales), worry, automaticity, routine, behavioral inhibition and activation.

Using an iterative model fitting procedure[91], we excluded explanatory variables based on the posterior distribution of the variable selection parameters ($\rho$), supported by inspection of cluster-specific posterior means and their 95% credible intervals: all variables that included the overall mean and therefore did not contribute to the identification of clusters were excluded from subsequent models.

Priors were set to the default weakly informative priors implemented in *PReMiuM*[90]. Cluster membership was modeled using a Dirichlet process mixture with a Gamma(1,1) prior on the concentration parameter. Cluster-specific parameters were assigned conjugate priors as implemented in *PReMiuM*, with Normal priors for cluster means and Inverse-Gamma priors for variances. Variable selection probabilities followed Beta(1,1) priors, corresponding to uniform prior belief. An initial burn-in of 50,000 sweeps was eliminated, and 100,000 sweeps retained from the Markov Chain Monte Carlo simulations from the posterior distribution of all model parameters. Additional explorative analyses of the moderating role of cluster membership on the association between ERN and self-control in daily life were conducted using GLMs ((11) *self-control failures ~ ERN + ERN:cluster membership*; (12) *self-control failures ~ ERN + ERN:cluster membership:impulsivity*).

### Reporting summary
Further information on research design is available in the Nature Portfolio Reporting Summary linked to this article.

### Results
We recruited 253 participants (after exclusions: n = 221, 46.6% female, mean age = 25.16 years, see "Methods" and Supplementary Tables 1–3 and Supplementary Figs. 1 and 2 for more detailed sample characteristics) along the dimensions of self-reported compulsivity[58] and impulsivity[60]. Participants completed questionnaires assessing psychopathological symptoms and personality traits associated with goal-directed control, including compulsivity and impulsivity (Fig. 2A). They also performed an arrow Flanker task to assess the ERN in a potential gain or loss context while the EEG was recorded (Fig. 2B, C), and completed an ecological momentary assessment of self-control in daily life over a 7-day period (Fig. 2D). See Fig. 2 for descriptive distributions of behavioral and neural measures.

To test our hypotheses, we applied logistic mixed-effects models to account for the nested data structure of the EMA assessments, as well as generalized linear models to analyze other outcome variables. Significant interaction effects were further explored using simple slopes analysis. Self-control was examined using all reported situations, with enactment serving as the outcome for the within-person dynamics analysis. Effects on desire and conflict reporting were also examined to disentangle this further. To maintain consistency with previous studies and ensure comparability[20], we additionally used the proportion of enacted conflict-laden desires relative to all reported answered prompts as a composite measure of self-control failures.

### Compulsivity, but not impulsivity, is significantly positively associated with desire enactment and self-control
First, we examined the influence of impulsivity and compulsivity on self-control and its within-person dynamics. We fitted a logistic mixed-effects

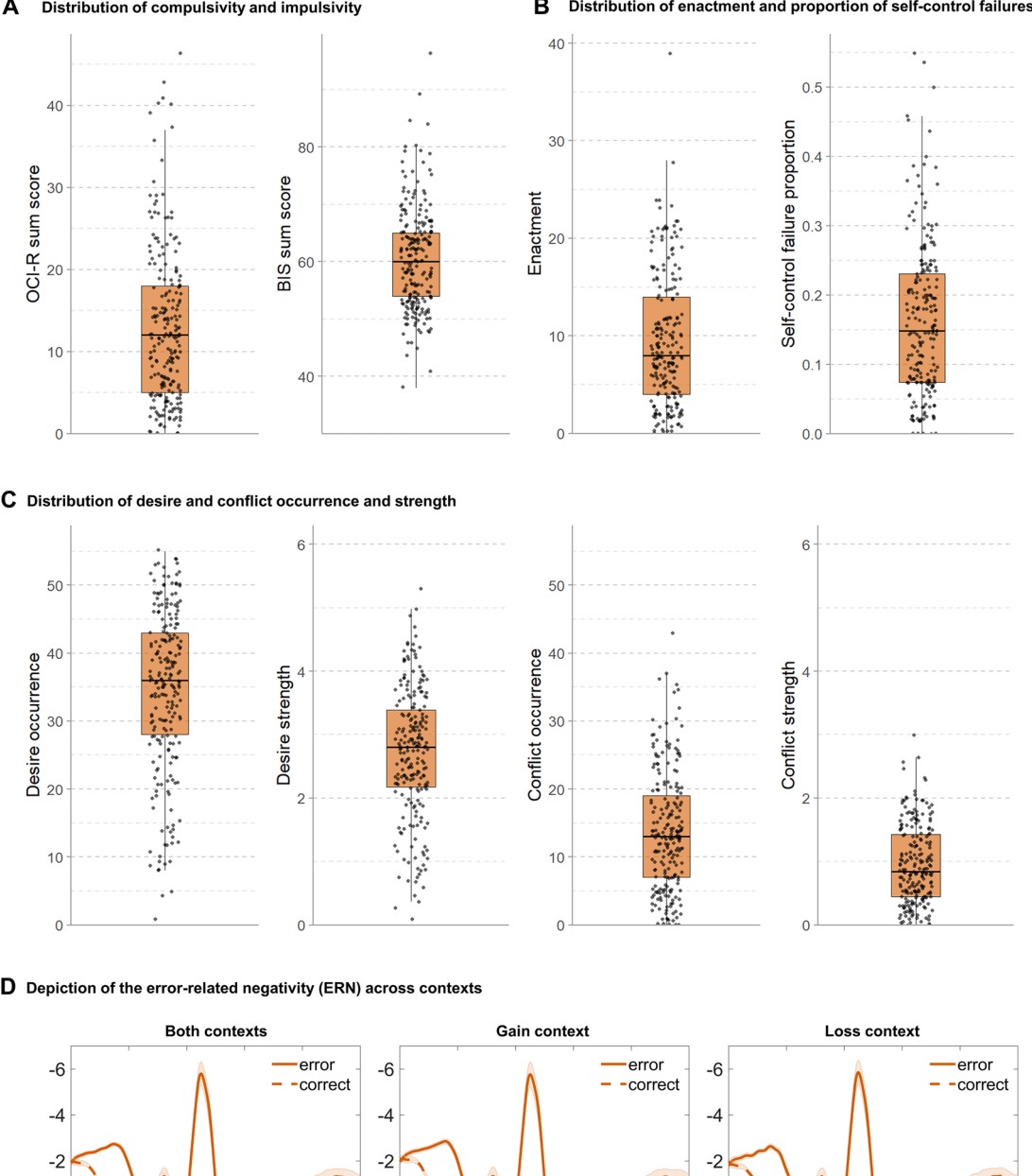

**Fig. 2 | Descriptive distributions of behavioral and neural measures.**
**A–C** Boxplots depicting the median and interquartile range for each variable; whiskers indicate 1.5 × IQR. Individual observations are displayed as jittered dots to illustrate the underlying data distribution. **A** Distribution of compulsivity and impulsivity scores as measured by the Obsessive-Compulsive Inventory-Revised (OCI-R; theoretical sum score range 0–72) and the Barratt Impulsiveness Scale (BIS-11; theoretical sum score range 30–120). **B** Distribution of desire enactment and the proportion of enacted conflict-laden desires relative to all reported desires as a composite measure of self-control failures. **C** Distribution of desire occurrence, mean desire strength, conflict occurrence, and mean conflict strength. **D** Response-locked event-related potentials (ERPs) for both contexts, gain, and loss contexts, showing the time course of incongruent error and correct trials averaged across Fz, FCz, and Cz. Shaded areas represent the SEM between subjects. Topographical maps depict mean response-locked activity at the mean ERN latency (54 ± 10 ms), shown separately for error and error-correct trials. ERN error-related negativity. All plots based on the complete sample (*n* = 221).

**Fig. 3 | The effects of impulsivity and compulsivity on desire enactment.** Results from the logistic mixed-effects model analyzing the influence of desire strength, conflict strength, impulsivity, and compulsivity on desire enactment ($n = 221$). Higher desire strength and higher compulsivity predicted more enactment, while higher conflict strength predicted lower enactment. Shaded areas represent the 95% confidence interval around the estimated effects. **A** Impulsivity had no significant effect on enactment ($\beta = -0.03$, 95% CI [$-0.77$, 0.71], $p = 0.938$). **B** Compulsivity significantly predicted enactment ($\beta = 0.31$, 95% CI [0.06, 0.56], $p = 0.016$). **C** Model-predicted interaction between desire strength and impulsivity and compulsivity on enactment. Desire strength had a significant positive effect ($\beta = 5.75$, 95% CI [5.27, 6.24], $p < 0.001$). The interaction effect with impulsivity and compulsivity was significant ($\beta = -0.67$, 95% CI [$-1.12$, $-0.23$], $p = 0.003$). **D** Model-predicted interaction between conflict strength and impulsivity and compulsivity on enactment. Conflict strength had a significant negative effect ($\beta = -1.83$, 95% CI [$-2.08$, $-1.57$], $p < 0.001$). The interaction effect with impulsivity and compulsivity was significant ($\beta = 0.41$, 95% CI [0.18, 0.65], $p < 0.001$). **C, D** Simple slopes depicting the moderating effect of compulsivity on the association between desire and conflict strength and desire enactment at different levels of impulsivity. First panel, simple slopes for the association for low ($-1$ SD below the mean), moderate (mean), and high ($+1$ SD above the mean) levels of compulsivity, in low ($-1$ SD below the mean) impulsivity. Second panel, simple slopes for the association for low ($-1$ SD below the mean), moderate (mean), and high ($+1$ SD above the mean) levels of compulsivity, in moderate (mean) impulsivity. Third panel, simple slopes for the association for low ($-1$ SD below the mean), moderate (mean), and high ($+1$ SD above the mean) levels of compulsivity, in high ($+1$ SD above the mean) impulsivity.

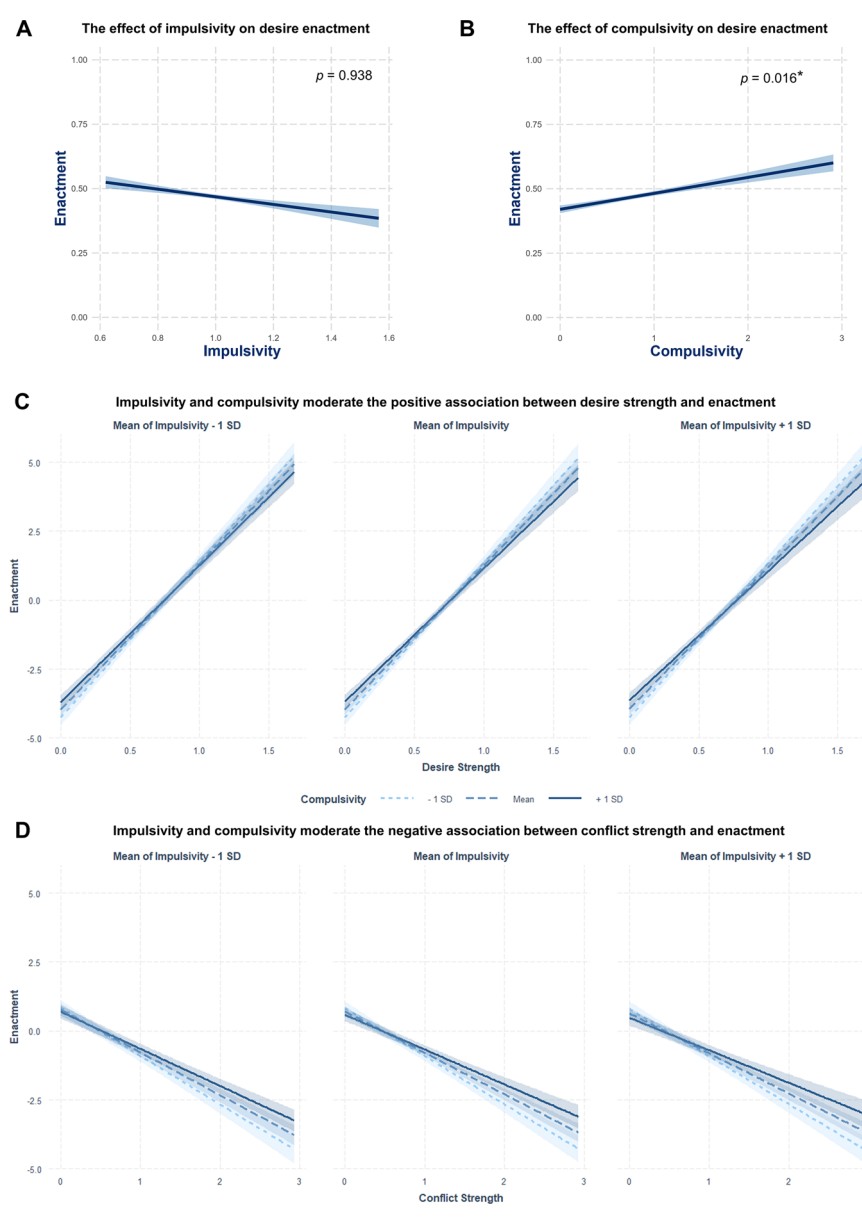

model predicting desire enactment from desire strength, conflict strength, impulsivity, and compulsivity. The model included random effects for desire strength and conflict strength, and interaction terms between these variables and both impulsivity and compulsivity. As shown in Supplementary Table 4, desire strength had a significant positive effect on desire enactment ($\beta = 5.75$, 95% CI [5.27, 6.24], $p < 0.001$), while conflict strength had a significant negative effect ($\beta = -1.83$, 95% CI [$-2.08$, $-1.57$], $p < 0.001$). There was no statistically significant association between impulsivity and enactment ($\beta = -0.03$, 95% CI [$-0.77$, 0.71], $p = 0.938$), whereas compulsivity had a significant positive effect ($\beta = 0.31$, 95% CI [0.06, 0.56], $p = 0.016$). Both the interaction between desire strength and impulsivity and compulsivity ($\beta = -0.67$, 95% CI [$-1.12$, $-0.23$], $p = 0.003$) and the interaction between conflict strength and impulsivity and compulsivity ($\beta = 0.41$, 95% CI [0.18, 0.65], $p < 0.001$) were significant. See Fig. 3 for a visualization of effects.

Simple slope analyses revealed that at higher levels of impulsivity, elevated compulsivity further weakened the influence of desire strength and conflict strength on enactment. All slopes were significant. These results suggest that both higher levels of impulsivity and compulsivity attenuate the associations between desire strength and conflict strength

with enactment. Similarly, the analysis of the proportion of enacted conflict-laden desires relative to all reported desires[20], showed that this was significantly predicted by compulsivity ($\beta = 0.04$, 95% CI [0.02, 0.07], $p < 0.001$), whereas the association with impulsivity did not reach significance ($\beta = 0.07$, 95% CI [$-0.03$, 0.17], $p = 0.164$). The inclusion of an interaction between impulsivity and compulsivity did not significantly improve prediction ($\Delta F(1, 217) = 0.01$, $p = 0.910$). See Supplementary Note 2.2 for detailed results.

These findings support the hypothesis that high compulsivity is associated with increased desire enactment or self-control failures (see Fig. 3A). In contrast, no evidence emerged for the assumed association between impulsivity and enactment (see Fig. 3B). Regarding the moderating effects, high impulsivity did not significantly amplify the influence of desire strength on enactment (hypothesis not confirmed, see Fig. 3C), but was associated with a reduced influence of conflict strength on enactment (hypothesis confirmed, see Fig. 3D). Conversely, high compulsivity was linked to a smaller influence of desire strength on enactment (hypothesis confirmed, see Fig. 3C), but also to a weaker association between conflict strength and enactment (hypothesis not confirmed, see Fig. 3D). See Supplementary Note 2.1 for detailed results.

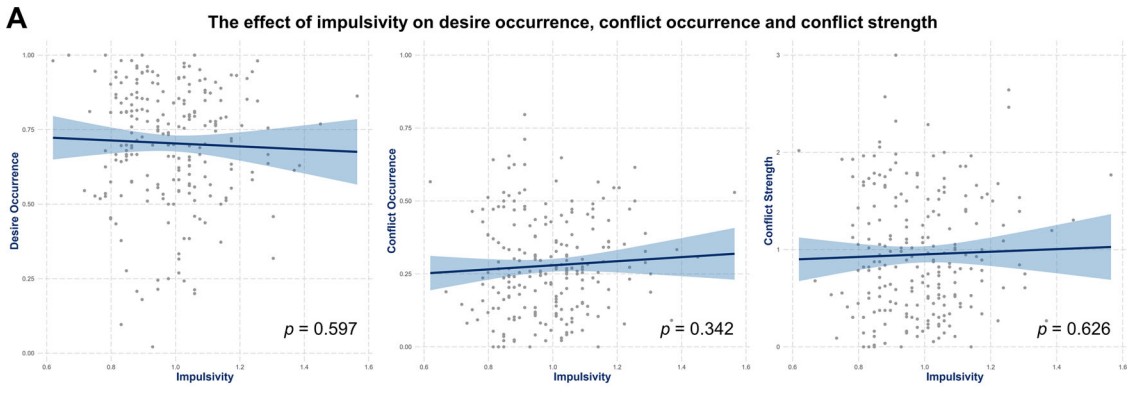

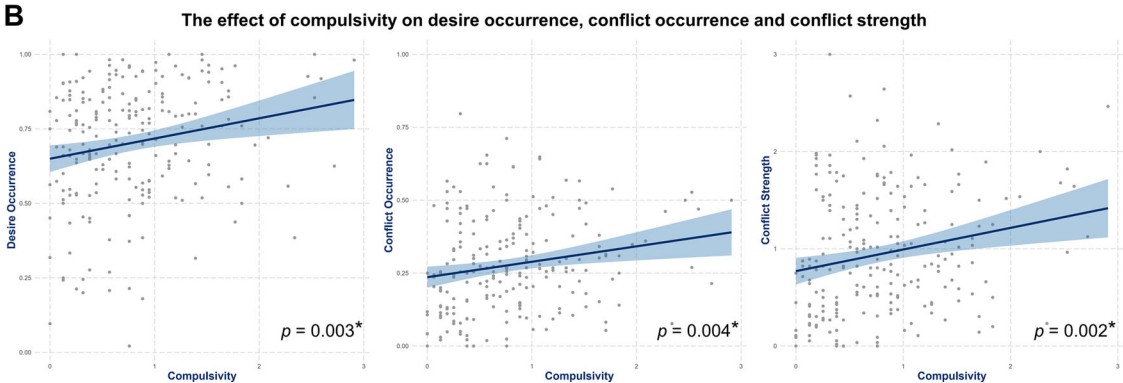

**Fig. 4 | The effects of impulsivity and compulsivity on desire occurrence and reported conflict.** Depiction of the effect of impulsivity on desire occurrence, conflict occurrence, and conflict strength ($n = 221$). Shaded areas represent the 95% confidence interval around the estimated effects. **A** There were no significant effects of impulsivity (desire occurrence: $\beta = -0.05$, 95% CI [$-0.23$, 0.13], $p = 0.597$); conflict occurrence: $\beta = 0.07$, 95% CI [$-0.08$, 0.22], $p = 0.342$; conflict strength: $\beta = 0.14$, 95% CI [$-0.42$, 0.69], $p = 0.626$. **B** Depiction of the effect of Compulsivity on desire occurrence, conflict occurrence, and conflict strength. Compulsivity significantly predicted desire occurrence ($\beta = 0.07$, 95% CI [0.02, 0.12], $p = 0.003$) and both conflict reporting ($\beta = 0.05$, 95% CI [0.02, 0.09], $p = 0.004$) and conflict strength ($\beta = 0.22$, 95% CI [0.09, 0.36], $p = 0.002$).

## Compulsivity, but not impulsivity, is significantly positively associated with desire occurrence and reported conflict

Second, we investigated how impulsivity and compulsivity influenced the reporting of desire and conflict. Using generalized linear models, we examined the associations of impulsivity and compulsivity with desire occurrence, reported conflict, and conflict strength. Compulsivity significantly predicted desire occurrence ($\beta = 0.07$, 95% CI [0.02, 0.12], $p = 0.003$), conflict occurrence ($\beta = 0.05$, 95% CI [0.02, 0.09], $p = 0.004$), and conflict strength ($\beta = 0.22$, 95% CI [0.09, 0.36], $p = 0.002$). Impulsivity showed no significant effects on desire occurrence ($\beta = -0.05$, 95% CI [$-0.23$, 0.13], $p = 0.597$), conflict occurrence ($\beta = 0.07$, 95% CI [$-0.08$, 0.22], $p = 0.342$), or conflict strength ($\beta = 0.14$, 95% CI [$-0.42$, 0.69], $p = 0.626$), and the interaction between impulsivity and compulsivity did not provide additional explanatory value (desire occurrence: $\Delta F(1, 217) = 0.13$, $p = 0.723$; conflict occurrence: $\Delta F(1, 217) = 0.29$, $p = 0.589$; conflict strength: $\Delta F(1,217) = 0.01$, $p = 0.948$). See Supplementary Note 2.3 for detailed results. Thus, our data provide no evidence that impulsivity was associated with the reporting of desire occurrence, conflict occurrence, or conflict strength (see Fig. 4A), whereas higher compulsivity was associated with these outcomes (see Fig. 4B).

## The association between the ERN and self-control is diminished in high compulsivity and impulsivity

Third, we examined the moderating influence of impulsivity and compulsivity on the previously established association between the error-related brain activity and self-control[19,20], given prior evidence of altered performance monitoring in disorders characterized by high levels of impulsivity and/or compulsivity. We fitted logistic mixed-effects models to predict desire enactment using desire strength, conflict strength, the ERN, impulsivity, and compulsivity, including desire and conflict strength as random effects.

In the base model (see Supplementary Table 5), desire strength positively predicted enactment ($\beta = 5.22$, 95% CI [4.91, 5.52], $p < 0.001$), while conflict strength had a significant negative effect ($\beta = -1.49$, 95% CI [$-1.65$, $-1.33$], $p < 0.001$). The ERN showed a non-significant effect ($\beta = 0.11$, 95% CI [$-0.08$, 0.31], $p = 0.258$). However, in the full model including an interaction of the ERN with impulsivity and compulsivity (see Supplementary Table 6), the ERN showed a significant positive association with enactment ($\beta = 0.26$, 95% CI [0.02, 0.51], $p = 0.033$). The three-way interaction between ERN, impulsivity, and compulsivity was significant and negative ($\beta = -0.20$, 95% CI [$-0.38$, $-0.01$], $p = 0.034$). Effects of desire strength ($\beta = 5.22$, 95% CI [4.91, 5.52], $p < 0.001$) and conflict strength ($\beta = -1.50$, 95% CI [$-1.65$, $-1.34$], $p < 0.001$) on enactment remained significant and in the same direction as in the base model. The three-way interaction was small in magnitude but robust. Parametric bootstrapping (2000 samples) yielded 95% confidence intervals closely matching Wald-based estimates ($\beta = -0.20$, 95% CI [$-0.38$, $-0.01$]), indicating a stable sampling distribution. Additional robustness checks, including refitting with multiple optimizers and subset-perturbation analyses, confirmed that the interaction remained consistently negative and of similar magnitude (see Supplementary Note 3.2 and Supplementary Table 7 for detailed results).

Simple slopes analyses examining the three-way interaction revealed that at low levels of impulsivity and compulsivity, higher (more negative) ERN amplitudes significantly predicted lower probability of enactment, consistent with greater self-control. This association weakened and was no longer significant for higher levels of impulsivity and compulsivity (see Fig. 5). See Supplementary Note 3.3 for detailed results. The effects were

**Fig. 5 | The effects of impulsivity and compulsivity on the association of ERN and desire enactment.** **A** Simple slopes for the moderating effect of impulsivity on the association between the error-related negativity (ERN) and desire enactment at different levels of compulsivity ($n = 221$). First panel, simple slopes for the association between ERN amplitude and enactment for low ($-1$ SD below the mean: $\beta = 0.23$, SE = 0.11, 95% CI [0.01, 0.45]), moderate (mean: $\beta = 0.22$, SE = 0.11, 95% CI [$-0.01$, 0.44]), and high ($+1$ SD above the mean: $\beta = 0.21$, SE = 0.11, 95% CI [$-0.01$, 0.43]) levels of impulsivity, in low ($-1$ SD below the mean) compulsivity. Second panel, simple slopes for the association between ERN amplitude and enactment for low ($-1$ SD below the mean: $\beta = 0.13$, SE = 0.10, 95% CI [$-0.07$, 0.33]), moderate (mean: $\beta = 0.10$, SE = 0.10, 95% CI [$-0.09$, 0.30]), and high ($+1$ SD above the mean: $\beta = 0.08$, SE = 0.10, 95% CI [$-0.12$, 0.28]) levels of impulsivity, in moderate (mean) compulsivity. Third panel, simple slopes for the association between ERN amplitude and enactment for low ($-1$ SD below the mean: $\beta = 0.03$, SE = 0.11, 95% CI [$-0.18$, 0.24]), moderate (mean: $\beta = -0.01$, SE = 0.12, 95% CI [$-0.24$, 0.22]), and high ($+1$ SD above the mean: $\beta = -0.05$, SE = 0.13, 95% CI [$-0.30$, 0.20]) levels of impulsivity, in high ($+1$ SD above the mean) compulsivity. **B** Simple slopes for the moderating effect of compulsivity on the association between the ERN and desire enactment at different levels of impulsivity ($n = 221$). First panel, simple slopes for the association between ERN amplitude and enactment for low ($-1$ SD below the mean: $\beta = 0.23$, SE = 0.11, 95% CI [0.01, 0.45]), moderate (mean: $\beta = 0.13$, SE = 0.10, 95% CI [$-0.07$, 0.33]), and high ($+1$ SD above the mean: $\beta = 0.03$, SE = 0.11, 95% CI [$-0.18$, 0.24]) levels of compulsivity, in low ($-1$ SD below the mean:) impulsivity. Second panel, simple slopes for the association between ERN amplitude and enactment for low ($-1$ SD below the mean: $\beta = 0.22$, SE = 0.11, 95% CI [$-0.01$, 0.44]), moderate (mean: $\beta = 0.10$, SE = 0.10, 95% CI [$-0.09$, 0.30]), and high ($+1$ SD above the mean: $\beta = -0.01$, SE = 0.12, 95% CI [$-0.24$, 0.22]) levels of compulsivity, in moderate (mean) impulsivity. Third panel, simple slopes for the association between ERN amplitude and enactment for low ($-1$ SD below the mean: $\beta = 0.21$, SE = 0.11, 95% CI [$-0.01$, 0.43]), moderate (mean: $\beta = 0.08$, SE = 0.10, 95% CI [$-0.12$, 0.28]), and high ($+1$ SD above the mean: $\beta = -0.05$, SE = 0.13, 95% CI [$-0.30$, 0.20]) levels of compulsivity, in high ($+1$ SD above the mean) impulsivity. Shaded areas represent the 95% confidence interval around the estimated effects.

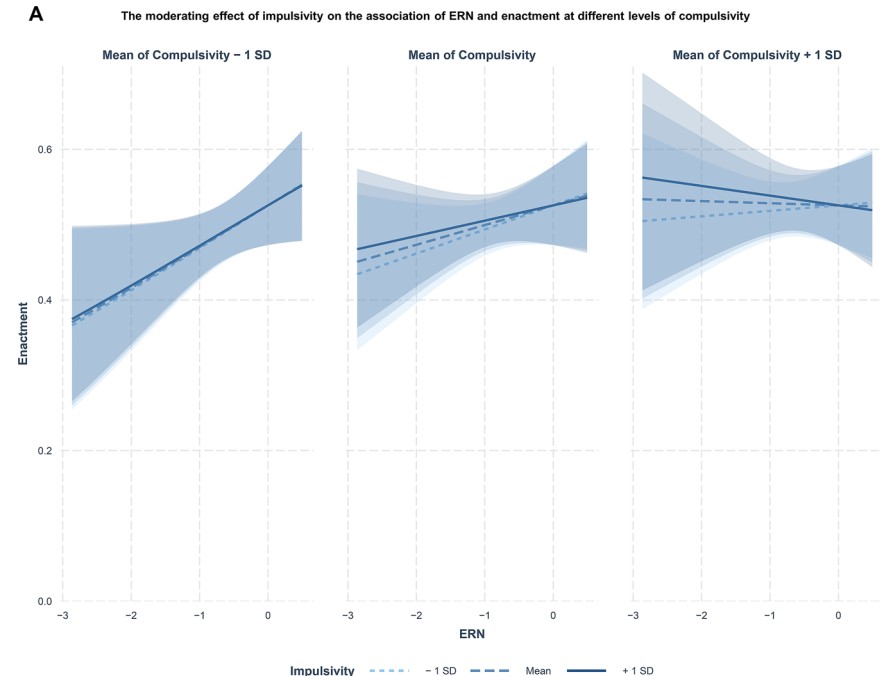

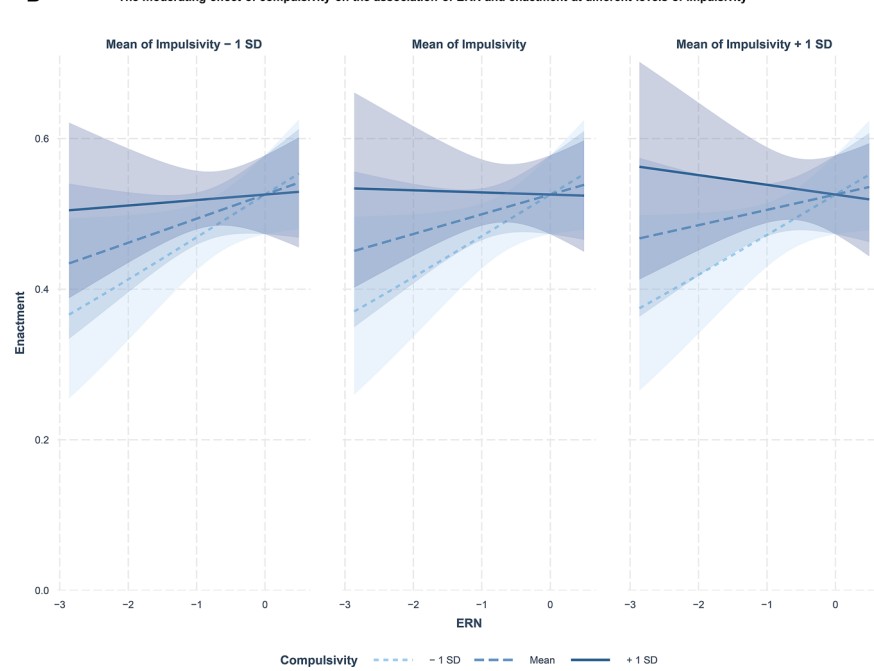

similar for both gain (interaction of ERN gain with impulsivity and compulsivity: $\beta = -0.19$, 95% CI [$-0.38$, $-0.01$], $p = 0.045$) and loss (interaction of ERN loss with impulsivity and compulsivity: $\beta = -0.20$, 95% CI [$-0.38$, $-0.02$], $p = 0.030$) motivational contexts (see Supplementary Note 3.4 and Supplementary Tables 8 and 9).

A similar pattern emerged when investigating the association between ERN and the proportion of enacted conflict-laden desires in all reported situations. The ERN significantly predicted self-control failures in daily life ($\beta = 0.06$, 95% CI [0.03, 0.09], $p = 0.001$), but only when its interaction with impulsivity and compulsivity was included ($\beta = -0.05$, 95% CI [$-0.08$, $-0.03$], $p < 0.001$). Again, this association was attenuated at higher levels of impulsivity and compulsivity (see Fig. S3 and Supplementary Note 3.5).

To summarize, these findings indicate that impulsivity and compulsivity interact in moderating the association between the ERN and self-control. Specifically, higher levels of impulsivity and compulsivity are associated with a diminished association between the ERN and self-control, independent of the motivational context. Detailed model outputs and robustness analyses are provided in Supplementary Notes 3.1–3.5. Primary fixed effects are summarized in Table 3.

**Cluster analysis reveals a compulsive/worry profile associated with increased self-control failures**

Fourth, we aimed to identify latent profiles that differ in self-control based on psychopathological symptoms and personality traits, beyond impulsivity and compulsivity. We used regression mixture modeling as an exploratory

**Table 3 | Primary fixed effects from mixed-effects models predicting enactment**

| The effects of impulsivity and compulsivity on desire enactment | | | |
|---|---|---|---|
| | **Enactment** | | |
| *Predictor* | **Odds ratio** | **95% CI** | **p** |
| Desire strength | 314.85 | 193.86–511.36 | **<0.001** |
| Conflict strength | 0.16 | 0.12–0.21 | **<0.001** |
| Impulsivity | 0.97 | 0.46–2.04 | 0.938 |
| Compulsivity | 1.36 | 1.06–1.75 | **0.016** |
| Desire strength × Impulsivity × Compulsivity | 0.51 | 0.32–0.80 | **0.003** |
| Conflict strength × Impulsivity × Compulsivity | 1.51 | 1.19–1.91 | **0.001** |
| **Moderation of the association between ERN and enactment** | | | |
| | **Enactment** | | |
| *Predictor* | **Odds ratio** | **95% CI** | **p** |
| Desire strength | 184.30 | 135.84 – 250.06 | **<0.001** |
| Conflict strength | 0.22 | 0.19–0.26 | **<0.001** |
| ERN | 1.30 | 1.02–1.66 | **0.033** |
| ERN × Impulsivity × Compulsivity | 0.82 | 0.68–0.99 | **0.034** |

Logistic mixed-effects models estimated using maximum likelihood (BOBYQUA optimizer) in the full sample ($n = 221$). Random intercepts and random slopes for desire strength and conflict strength were included at the participant level. Full model specifications and random effects are provided in Supplementary Tables 4–9.
Statistically significant $p$ values are shown in bold.

approach to examine this, using the proportion of enacted conflict-laden desires relative to all reported desires as a composite measure of self-control failures. In addition to impulsivity and compulsivity, we included four additional impulsivity facets (urgency, lack of premeditation, lack of perseverance, and sensation seeking), depression, three different scales assessing aspects of anxiety, worry, automaticity, and routine as facets of habitual propensity, as well as behavioral inhibition and activation as explanatory variables.

We determined that two clusters were optimal using the partitioning around medoids method. We then fitted a profile regression model to examine the effect of the predictor variables on self-control failures through cluster membership[90]. Variables that did not contribute to cluster identification were removed from the model: these included all facets of impulsivity (urgency, lack of premeditation, lack of perseverance, sensation seeking), depression, somatic and cognitive anxiety, automaticity and routine, behavioral inhibition, and activation.

The final model retained three variables (see Fig. 6): compulsivity, anxiety as measured by the Depression Anxiety Stress Scales (DASS-21), and worry (PSWQ). The first cluster ($n = 130$), characterized by lower levels of compulsivity, anxiety, and worry, reported fewer self-control failures (empirical mean = 0.142, posterior mean = 0.149, 95% highest posterior density interval (HPD) = 0.118–0.180). In contrast, the second cluster ($n = 91$), characterized by higher compulsivity, anxiety, and worry, reported more self-control failures (empirical mean = 0.188, posterior mean = 0.178, 95% HPD = 0.143–0.214). Sensitivity analyses were conducted under alternative prior assumptions on the within-cluster covariance structure of the covariates, including an independent Normal prior and a separation prior for the covariance matrix. This yielded highly similar posterior cluster separation estimates across specifications (see Supplementary Note 6.1 and Supplementary Table 10), indicating robustness to prior assumptions. A follow-up analysis confirmed that cluster membership significantly predicted self-control failures ($\beta = 0.04$, $p = 0.002$, $\beta_{standardized} = 0.21$, 95% CI [0.08, 0.34]; model: adjusted $R^2 = 0.04$, $F(1, 219) = 0.64$, $p = 0.002$). Thus, these findings

suggest that individuals with elevated measures of compulsivity, anxiety, and worrying may exhibit more self-control failures.

To exploratorily examine the latent profiles in relation to error processing, we examined whether cluster membership moderated the association between ERN amplitude and self-control failures. The ERN significantly predicted self-control failures in daily life ($\beta = 0.04$, 95% CI [0.01, 0.07], $p = 0.007$), but only when its interaction with Cluster membership ($\beta = -0.04$, 95% CI [−0.07, −0.01], $p = 0.008$) was included, indicating that the strength of the association between ERN and self-control failures differed between clusters. Specifically, in the cluster characterized by low compulsivity, anxiety, and worrying, larger ERN amplitudes were associated with fewer self-control failures, whereas this association was attenuated in the cluster characterized by high expressions of compulsivity, anxiety, and worrying. When further testing whether impulsivity modulated this relationship, a significant three-way interaction with impulsivity emerged only in the cluster characterized by high compulsivity, anxiety, and worrying ($\beta = -0.11$, 95% CI [−0.21, −0.01], $p = 0.034$), indicating that higher impulsivity further attenuated the association between ERN and self-control failures. In contrast, this interaction was not significant in the low symptom cluster ($\beta = -0.07$, 95% CI [−0.16, 0.03], $p = 0.177$). See Supplementary Note 6.2 for detailed results.

### Additional analyses regarding behavioral parameters

Post-error slowing, the reaction time (RT) on correct incongruent trials, and the RT on erroneous incongruent trials showed no significant association with self-control (all $p > 0.1$). Higher error rate was associated with lower self-control ($\beta = 0.06$, 95% CI [0.02, 0.10], $p = 0.001$), higher post-error accuracy ($\beta = -0.14$, 95% CI [−0.21, −0.07], $p < 0.001$), and higher post-correct accuracy ($\beta = -0.12$, 95% CI [−0.20, −0.04], $p = 0.003$) predicted higher self-control in daily life. See Supplementary Notes 4.1 and 4.2 for detailed results.

## Discussion

We investigated how self-reported impulsivity and compulsivity relate to daily self-control and moderate the relationship between error-related brain activity (ERN) and self-control. As predicted, high compulsivity was associated with increased desire enactment and self-control failures. An exploratory cluster analysis further suggested that a profile characterized by elevated compulsivity, anxiety, and worry was associated with more frequent self-control failures. Contrary to expectations, we did not find evidence for a direct association between impulsivity and desire enactment or self-control failures, cluster identification, or desire and conflict occurrence and conflict strength.

Impulsivity and compulsivity also interacted in moderating self-control dynamics. Specifically, the positive relationship between desire strength and enactment was weakened at higher impulsivity, particularly when compulsivity was also high. Similarly, the negative association between conflict strength and desire enactment was diminished at higher levels of both traits. These patterns only partially aligned with our hypotheses, which anticipated more distinct and opposing effects for each trait. Furthermore, high compulsivity was linked to increased desire occurrence, extending beyond our initial hypothesis of increased conflict occurrence and strength.

Finally, consistent with our predictions, compulsivity and impulsivity interacted in moderating the relationship between ERN amplitudes and self-control. The association between ERN and self-control was significant only at low levels of both traits and diminished as impulsivity and compulsivity increased, with the strongest attenuation observed when both traits were elevated. This was supported by explorative cluster analyses showing that the association between ERN and self-control was only significant in the cluster characterized by low compulsivity, anxiety, and worry, and that impulsivity attenuated this association only in the cluster characterized by high symptom expression. Supporting the relevance of error processing for self-control, error rate, post-error, and post-correct accuracy all predicted self-control. These findings highlight the complex interplay between

**Fig. 6 | Bayesian profile regression on self-control.**
**A** Visualization of the model, including cluster characteristics (*n* = 221). Empirical data show the average response (Y) per cluster; Size indicates the number of participants in each cluster; Risk denotes the posterior distribution of the plausible range of self-control failures (Expected response "E[Y]") and is represented with the posterior mean and posterior interval estimates for each cluster. The explanatory variables compulsivity, anxiety (DASS-21; see Supplementary Note 1.2), and worry (PSWQ; see Supplementary Note 1.2) are shown with their posterior mean and posterior interval estimates for each cluster, bottom panels correspond to estimated posterior standard deviations; all values are standardized (z-scored) across participants. Colored dots represent the 5th and 95th percentiles of the posterior distribution for the particular variable within a cluster, while the black horizontal line represents the population mean for the variable of interest. Red indicates that the credible intervals at the 10th percentile do not overlap with the overall population mean, green indicates that the 90th or 10th percentile credible intervals overlap with the population mean and blue indicates that the 90th percentile credible intervals do not overlap with the population mean. **B** Posterior distribution of reported self-control in the two clusters (*n* = 221).

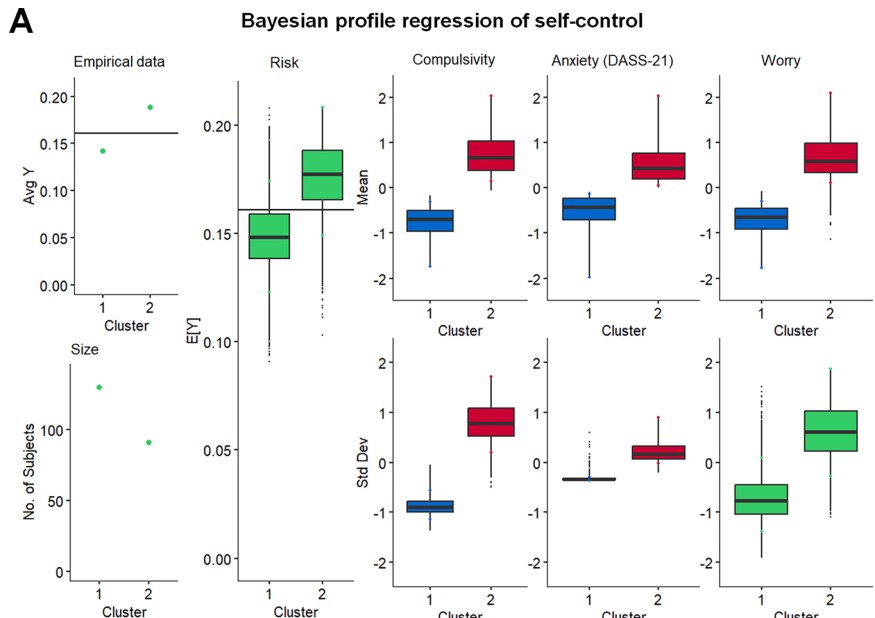

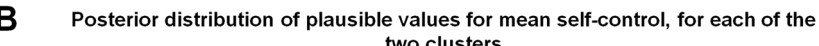

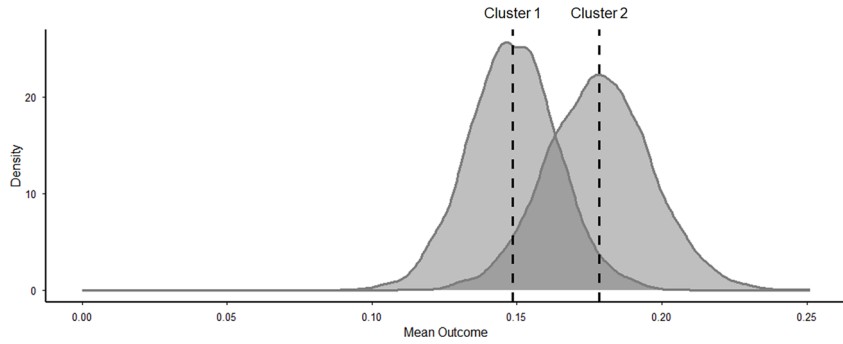

impulsivity, compulsivity, and neural correlates of performance monitoring, and underscore the importance of considering trait interactions when investigating self-control and its underlying cognitive and neural processes.

The observed association between high compulsivity and deficits in self-control aligns with research linking compulsivity to rigid goal-pursuit and reduced flexibility in the employment of cognitive control[92,93]. Accordingly, our findings align with previous studies showing that individuals with OCD often exhibit a strong desire for control yet a diminished sense of actual control, manifesting as increased conflict occurrence and strength[45,94]. Furthermore, individuals high in compulsivity may be more aware of their desires and conflicts, which could enhance their monitoring of such experiences, reflected in elevated ERN amplitudes[95,96]. This may also be true for individuals high in anxiety and worry[97,98]. Importantly, our findings suggest that heightened performance monitoring does not necessarily translate into more effective self-control. This points to a potential decoupling between monitoring and control implementation, possibly due to a failure in utilizing the monitoring signal adaptively. Similar dissociations have been observed in individuals with low action orientation, who show enhanced ERN amplitudes but fail to recruit control mechanisms and instead respond with rumination and negative affect[55,99]. Conversely, the association between impulsivity and self-control did not reach statistical significance in the present analyses. Although a true null effect cannot be ruled out, this finding contrasts with common assumptions about the role of impulsivity in self-control[4]. Impulsivity has been linked to reduced ERN amplitudes and altered performance monitoring in prior work[100], suggesting impaired error and conflict monitoring. Moreover, individual differences in

goal representation, valuation, or conflict and temptation appraisal may shape how self-control is experienced and reported[14,51]. Further research is required to clarify how these processes relate to everyday self-control dynamics. Both high impulsivity and compulsivity attenuated the influence of desire and conflict strength on desire enactment. For compulsivity, this suggests that rigid and automatic behaviors dominate, reducing adaptation to internal motivational states. High impulsivity, in contrast, may be associated with reduced integration of situational factors and limited deliberation, resulting in more reactive behaviors. Thus, while compulsivity promotes rigidity, impulsivity may increase behavioral reactivity due to diminished contextual modulation. Together, these traits may reduce sensitivity to internal and external contextual factors, shifting behavior toward increased rigidity and reactivity.

We found evidence that compulsivity and impulsivity interact to moderate the association between the ERN and self-control. Specifically, the predictive value of the ERN amplitude was attenuated at high levels of both traits. This suggests that while compulsive individuals may show heightened monitoring, as reflected by larger ERN amplitudes, this signal has diminished behavioral relevance when impulsivity is also high. Conversely, impulsivity only moderates the association between ERN and self-control when compulsivity is elevated. These findings extend existing frameworks by demonstrating that impulsivity and compulsivity not only interact to shape daily self-control dynamics but also modulate the functional relevance of neural measures of performance monitoring for the employment of control in daily life, irrespective of the motivational context of performance monitoring.

We observed evidence for reduced context sensitivity in individuals scoring high on impulsivity and compulsivity, as reported in Overmeyer and Endrass[63]. These trait effects may reflect reduced flexibility in the performance monitoring system, together with attenuated adaptation of the ERN across motivational contexts[33,63]. Since the ERN is linked to the motivational significance of errors[33,101], an inflexible performance monitoring system that fails to distinguish the motivational significance between different events may provide less useful information for guiding behavior. As a result, performance monitoring may exert diminished influence on action selection in these individuals.

Alternatively, the attenuated association between performance monitoring and self-control may reflect downstream deficits in utilizing performance monitoring signals. Prior research suggests that even when individuals detect conflict or errors (as reflected in larger ERN amplitudes), they may fail to mobilize control resources accordingly, especially under high negative affect or low action orientation[55,99,102]. Deficits in the cognitive control network[17,18] or altered functioning in the valuation network[13,14] may contribute. When control or long-term goals are assigned low subjective value, monitoring signals may fail to trigger adaptive adjustments. A further explanation for the attenuated association in highly impulsive individuals may be a lack of motivation to recruit and implement cognitive control[103]. This might also reflect a decoupling of "liking" and "wanting," in which the motivation ("wanting") may become disconnected from the actual pleasure ("liking") derived from a certain stimulus or behavior[28,104]. In mental disorders characterized by impaired self-control, this disconnection may drive individuals to pursue certain behaviors, or rewards, such as substances, not only despite the fact that they are in conflict with their long-term goals, but also regardless of the pleasure they derive from them[28]. Consequently, these actions may persist despite the absence of both goal alignment and satisfaction, highlighting a disruption in self-control mechanisms. In summary, individuals with high levels of impulsivity and compulsivity may have difficulty using the information from the performance monitoring system due to reduced flexibility and informational value, misalignment between personal goals and those assessed through our interventive self-control measures, or diminished motivation to use the information provided.

## Limitations
Our cross-sectional design limits causal inferences about these relationships, and future longitudinal studies are needed to examine the temporal dynamics of trait-state interaction. Including clinical populations and examining additional moderators, such as psychopathological symptoms or stress, will also be important to refine models of adaptive behavior and inform interventions. Our measure of self-control also focused exclusively on interventive self-control[4]. Importantly, while our study focused on interventive self-control, i.e., the mobilization of control in the presence of a current temptation[4], further work should also examine preventive self-control and strategic goal pursuit[5,105] to capture the full spectrum of self-control processes. Although our flanker task included external feedback, its potential interaction with internal error monitoring is likely minimal. Since incorrect responses were always followed by negative feedback, rendering it relatively uninformative, the robust ERN observed in our data likely reflects internal error processing, in line with the first indicator hypothesis[106]. In addition, while we have demonstrated that the connection between the ERN and self-control in daily life is moderated by impulsivity and compulsivity, these effects are small and should be further examined in clinical samples. For example, the relationship between impulsivity and self-control may vary depending on the presence and severity of substance use. Recent evidence suggests that using measures for error monitoring that integrate information from EEG and functional magnetic resonance imaging may be helpful in improving predictive value for mental health outcomes[107].

## Conclusion
Taken together, our results suggest that individuals high in compulsivity and anxiety report more desire enactment and failures in self-control, report more desires and experience more conflict about these desires, and show reduced ability to translate performance monitoring into effective self-control. In contrast, we did not find evidence for a direct association between impulsivity and self-reported daily self-control. However, impulsivity interacted with compulsivity in attenuating the association between the ERN and self-control. These findings point to a decoupling between performance monitoring and the implementation of control in daily life in individuals with poor goal-directed control. Our findings underscore the importance of considering interactions between performance monitoring, motivational systems, and trait-level self-regulation in understanding self-control.

## Data availability
The data that support the findings of this study are available in the Open Science Framework with the identifier https://doi.org/10.17605/OSF.IO/4GPND.

## Code availability
The code that supports the findings of this study is available in the Open Science Framework with the identifier https://doi.org/10.17605/OSF.IO/4GPND. This includes the scripts used for analyzing the dataset reported in the manuscript.

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

## Acknowledgements

This work was supported by the collaborative research centers CRC 940 and CRC/TRR 265, funded by the Deutsche Forschungsgemeinschaft (DFG, German Research Foundation). The funders had no role in study design, data collection and analysis, decision to publish, or preparation of the manuscript. Computing time for data analysis was provided by the Center for Information Services and High Performance Computing (ZIH) at Dresden University of Technology. The authors thank Julia Berghäuser and their student research assistants for their contributions to data collection. The authors also thank all participants for their time.

## Author contributions

The following list of author contributions is based on the CRediT taxonomy. Rebecca Overmeyer: conceptualization; methodology; software; validation; formal analysis; investigation; data curation; writing—original draft; writing—review and editing; visualization; project administration. Anja Kräplin: methodology; writing—review and editing; supervision. Thomas Goschke: conceptualization; methodology; writing—review and editing; funding acquisition. Tanja Endrass: conceptualization; methodology; software; validation; resources; writing—review and editing; supervision; project administration; funding acquisition.

## Funding

## Competing interests

The authors declare no competing interests.
