## [Transparent Peer Review file · Communications Psychology]

The association between the error-related negativity and self-control is moderated by impulsivity and compulsivity

Corresponding Author: Dr Rebecca Overmeyer

Version 0:

Decision Letter:

Dear Dr Overmeyer,

Thank you for submitting your manuscript titled "THE ASSOCIATION BETWEEN THE ERROR-RELATED NEGATIVITY AND SELF-CONTROL IS MODERATED BY IMPULSIVITY AND COMPULSIVITY" to Communications Psychology. We have given the paper our careful consideration and find it of potential interest. However, due to certain shortcomings we are concerned that sending the current manuscript out to review could lead to unnecessary delays and quite possibly an undesirable outcome of the review process.

In particular, we ask that the description of the hypotheses in the manuscript maps exactly onto their description in the preregistration document. Ideally, hypotheses are presented verbatim the source text. You may also wish to improve the anonymisation of your manuscript by removing the name of the ethics-granting institution.

We would therefore like to invite you to revise your manuscript to address these concerns before we make a final determination on whether to send your manuscript for external review.

We shall hope to receive your revised version as soon as you are able to complete the suggested revisions. If something similar is published in the interim we will have to consider the impact it has on the novelty of a revised manuscript.

If you anticipate a delay of more than four weeks, please let us know. Should your manuscript be substantially delayed without notifying us in advance and your article is eventually published, the received date may be that of the revised, not the original, version.

We also ask that you ensure your manuscript complies with our editorial policies and reporting requirements.

To that end, we require revised manuscripts to be accompanied by a completed item: a reporting summary that collects information on study design and procedure.

- <https://www.nature.com/documents/nr-reporting-summary.pdf>>Nature Research Reporting Summary

Your revised manuscript can only be sent to referees if the checklist is completed and uploaded with the revision.

If you are not interested in submitting a suitably revised manuscript in the future please let me know immediately so we can close your file. If you have any questions, please contact me.

Please use the link below when you are prepared to resubmit.

Link Redacted

Thank you for your interest in Communications Psychology.

Best regards,
Marika Schiffer

Marika Schiffer, PhD
Chief Editor
Communications Psychology

Version 1:

Decision Letter:

Dear Dr Overmeyer,

Thank you for your patience during the peer-review process. Your manuscript titled "THE ASSOCIATION BETWEEN THE ERROR-RELATED NEGATIVITY AND SELF-CONTROL IS MODERATED BY IMPULSIVITY AND COMPULSIVITY" has now been seen by 2 reviewers, and I include their comments at the end of this message. They find your work of interest but raised some important points. We are interested in the possibility of publishing your study in Communications Psychology, but would like to consider your responses to these concerns and assess a revised manuscript before we make a final decision on publication.

We therefore invite you to revise and resubmit your manuscript, along with a point-by-point response to the reviewers. Please highlight all changes in the manuscript text file.

Editorially, we consider the following issues critical:

The reporting of the preregistration and of deviations from it needs to be further improved. Please refer to the attached checklist. Presentationally, the work would also benefit from display items that show the data more descriptively. Please also attend closely to our guidance on reporting and interpretation of non-significant findings, as manuscripts are frequently returned to authors when revisions do not comply.

As you revise the manuscript in response to these issues, please also implement all requests in the attached Mandatory Revision Requests document. All requirements listed in this document need to be fully met, or the work will be returned to you for further revisions without peer review. This workflow is in place to increase the likelihood that the paper will be accepted for publication. It reduces the number of rounds of revision (and review) and ensures that the reviewers vet a version of the article that is compliant with journal policies. If you have any questions regarding the required revisions, please contact the journal prior to resubmission to avoid a negative outcome.

Please submit the following items:

- Revised manuscript
- Point-by-point response to the referees' comments
- Mandatory Revision Requests Table (attached).
- Cover letter (as a separate document)

via this link: Link Redacted .

**** This url links to your confidential home page and associated information about manuscripts you may have submitted or are reviewing for us. If you wish to forward this email to co-authors, please delete the link to your homepage first ****

Best regards,

Marika

REVIEWER EXPERTISE:

Reviewer #1: cognitive control, EEG

Reviewer #2: cognitive control, EEG

REVIEWER REPORTS:

Reviewer #1 (Remarks to the Author):

The manuscript examines how impulsivity and compulsivity influence daily self-control and moderate the association between the error-related negativity (ERN) and self-control. The study is well powered ($N = 221$ after exclusions), preregistered, and employs an ambitious multi-method design combining EEG, ecological momentary assessment (EMA), and questionnaires. The main findings are that compulsivity, but not impulsivity, predicts more frequent self-control failures, and that the association between ERN and self-control is present only at low levels of both traits. The work is carefully executed and offers a valuable contribution.

Major issues

1. The introduction frequently discusses performance monitoring broadly, while the empirical analyses focus specifically on error monitoring (ERN). It would strengthen the paper to narrow the framing or explicitly note that only one aspect of performance monitoring is addressed.
2. Hypotheses center on EMA indices of desire, conflict, and enactment, yet the introduction provides relatively little theoretical or empirical context for these measures. More discussion of prior EMA research on self-control would give the hypotheses a stronger foundation.
3. The power analysis states that three predictive effects were the focus, but it is not clear which effects these were or on what empirical basis the assumed effect size ($f^2 = 0.05$) was chosen. Since no identical studies exist in the literature, a clearer justification is needed.
4. The flanker task provided both speed- and accuracy-related feedback. Prior work indicates that external correctness feedback can reduce the salience of internal error-monitoring signals. This potential issue should be discussed as a limitation when interpreting ERN results.
5. The three-way ERN \times impulsivity \times compulsivity interaction is statistically significant but likely small in effect size. More detail on robustness checks (e.g., bootstrapped confidence intervals, sensitivity analyses) would increase confidence that this effect is not fragile.
6. I have the impression that the manuscript doesn't explicitly say how ERN was entered into the models. Did the authors take each participant's mean ERN amplitude (e.g., averaged across all errors) and use this as the predictor? Or did they, for example, split by gain/loss context and then average? Did they consider trial-level ERN values at all?

Minor issues

1. The EEG was re-referenced to the average of all electrodes. While this approach is common, mastoid (or near-mastoid, e.g., TP9 & TP10) references are often preferred for midline components such as the ERN and CRN. The authors should justify their choice.
2. The ERN and CRN waveforms are canonical (Fig. S2), but providing an additional topographic map (for the ERN–CRN difference) would help confirm that the observed effects show the expected scalp distribution.
3. Please report the mean, minimum, and maximum number of trials per condition (after preprocessing and artifact rejection).
4. The discussion devotes substantial space to speculating about why impulsivity did not predict self-control. While some speculation is appropriate, this section could be tightened and balanced by acknowledging the possibility of genuinely null effects.
5. It would be useful to clarify how missing or skipped EMA prompts were handled in the analyses.
6. Since the cluster analysis is exploratory, it should be framed more as hypothesis-generating than as evidence of distinct subgroups.

Reviewer #3 (Remarks to the Author):

The authors examined the neural underpinnings of self-control as mediated by impulsivity and compulsivity. They measured various traits, as well as diary reports of daily self-control and ERP measures in a large sample. Results showed that compulsivity was associated with desire enactment, whereas impulsivity was not (although both had attenuating effects). Compulsivity also predicted the occurrence of desires/conflicts. With respect to the ERN, complex 3-way interactions suggested that at lower levels of impulsivity/compulsivity the ERN negatively predicted enactment on desires.

The work was preregistered, is super well powered for an EEG experiment and well worth publishing here. I have comments below that I hope will help improve the paper, but in general, I see them as relatively minor.

While the authors give a good (if standard) description of self-control in their introduction I think it could be more nuanced given some recent claims. First, self-control isn't directly measured - impulsivity, compulsivity and lack of perseverance/premeditation are taken as indicators of self-control. But, as the originators of the self control scale most commonly used in the literature (Tangney et al) have recently suggested, these things might better reflect "failures" of self-control rather than any specific underlying mechanism. (Schrader, S. W., & Tangney, J. P. (2025). Rethinking the Measurement of Self-Control: Distinguishing Among Self-Control Capacity, Urge Intensity, and Behavior Outcomes. *Basic and Applied Social Psychology*, 47(1), 37-46.) I see throughout that failures of self-control are the focus, but some treatment of how this is (poorly) measured might be warranted. A tripartite distinction between goal initiation, goal persistence and inhibitory control may be a more nuanced way of exploring self-control (I think initiation is missing here unless you assume premeditation to be a measure of this). I don't think this is damning to the current paper - I just think a little more nuance is warranted.

Second, recent work from Eisenberg et al (2019) suggests that surveys and not tasks like the flanker, do a better (although not dramatically better) job of predicting real-world outcomes of self-regulatory control issues. (Eisenberg, I. W., Bissett, P. G., Zeynep Enkavi, A., Li, J., MacKinnon, D. P., Marsch, L. A., & Poldrack, R. A. (2019). Uncovering the structure of self-regulation through data-driven ontology discovery. *Nature communications*, 10(1), 2319.) I think this is relevant here too. Do we even know if ERNs (to tasks, which don't predict well) are predictive of real-world self-regulatory challenges (i.e., do ERNs predict drug and alcohol misuse as just one example?). Again, this is not damning to this paper, but I think it is worth including in the discussion at least as a potential limitation of this (and much of the self-control) work. The everyday reports of self-control challenges get at this somewhat but are not quite as direct as asking whether ERNs (vs. trait metrics collected) do a better (or even equivalent) job of predicting outcomes like substance abuse (which was prominently mentioned in the introduction).

With respect to the cluster analysis, did cluster 1 report fewer "failures" or just fewer situations demanding self-control overall? I know the measure used is a composite, but composites, like difference scores, hide somewhat the contribution of things like base rate of actual conflicts experienced.

Also, I was expecting some exploration of how these distinct clusters might then relate to the ERN amplitude? Seems a glaring oversight to not even discuss this.

Minor points:

line 54: SUD (I presume substance use disorder?) is not spelled out - probably don't need to spell out OCD, but is SUD really as highly used and known? In general, acronyms help the writer far more than the reader, so I would avoid them wherever possible. Turns out, you spell it out later in the Introduction - just move that forward to first usage (or eliminate the acronym altogether!).

The introduction in general puts forward a lot of causal models (e.g., higher ERN=higher response to errors=leads to increased demand for self-control="thereby producing more effective self-control" as just one example). Do we really have the data to speak to such causal arrows? I might temper this language to be about "relations" as opposed to "leads to" type language. In particular, at many time points it reads like the ERN causes self-control challenges - I'm not sure that's what the authors really want to suggest is it?

line 188: I don't think an effect of $p=0.258$ should be talked about as a "non-significant positive effect" - it's just plain ol' non-significant at that point, no?

Not sure how viable this is (some LLM analysis perhaps given the size of the data set) - but I was curious as to the nature of the conflicting desires that participants reported experiencing in the 7-day period. In part, this is to determine the "seriousness" of the conflicts - if they were relatively minor (as I suspect they would be in this sample) then we might expect more dramatic effects in pathological populations like obsessive-compulsive or substance use disorders. Be worth briefly commenting on.

* TRANSPARENT PEER REVIEW: Communications Psychology uses a transparent peer review system. This means that we publish the editorial decision letters including Reviewers' comments to the authors and the author rebuttal letters online as a supplementary peer review file. However, on author request, confidential information and data can be removed from the

published reviewer reports and rebuttal letters prior to publication. If your manuscript has been previously reviewed at another journal, those Reviewers' comments would not form part of the published peer review file.

Version 2:

Decision Letter:

Dear Dr Overmeyer,

Your manuscript titled "THE ASSOCIATION BETWEEN THE ERROR-RELATED NEGATIVITY AND SELF-CONTROL IS MODERATED BY IMPULSIVITY AND COMPULSIVITY" has now been seen by our reviewers, whose comments appear below. In light of their advice I am delighted to say that we are happy, in principle, to publish a suitably revised version in Communications Psychology.

We therefore invite you to revise your paper one last time to address the remaining list of editorial requests. At the same time we ask that you edit your manuscript to comply with our format requirements and to maximise the accessibility and therefore the impact of your work.

EDITORIAL REQUESTS:

SUBMISSION INFORMATION:

OPEN ACCESS:

Link Redacted

Best regards,

Marieke

Marieke Schiffer, PhD
Chief Editor
Communications Psychology

REVIEWERS' COMMENTS:

Reviewer #1 (Remarks to the Author):

The authors have addressed my previous concerns carefully, and I appreciate the additional robustness analyses and methodological clarifications. I have two remaining minor requests prior to acceptance, concerning (1) the strength of interpretation in the Abstract, and (2) the color scheme used for the ERN topographies.

(1) In particular, statements such as “These findings demonstrate that impulsivity and compulsivity jointly reduce the behavioral relevance of performance monitoring” and “underscoring the importance of accounting for trait interactions into cognitive and neural models of self-control” (Abstract) appear somewhat overconfident given the correlational design and the modest size of the three-way interaction. I recommend replacing such phrasing with more cautious language and explicitly acknowledging that mechanistic interpretations remain speculative.

(2) In Fig. 1C and Fig. 2D, the ERN topographies are displayed using a color scheme in which negative values are shown in red and positive values in blue. This is opposite to common ERP conventions, where negative amplitudes are typically depicted in blue and positive in red, and may therefore be misleading, especially given that the ERN is a negative-going component. I recommend either adopting a conventional color mapping or, at minimum, making the color scale more prominent (e.g., larger values / clearer labeling) to ensure that the polarity assignment is immediately apparent to the reader.

Reviewer #3 (Remarks to the Author):

The authors have done a good job addressing my initial concerns. I think the piece is ready for publication.

Response Letter:

First of all, we thank the reviewers for their comments and suggestions for revising our manuscript. In the following, we will comment on the points and arguments made by the reviewers. The reviewers' comments are included in the response letter combined with the responses to the comments (in italic print for clarity). Changes in the manuscript are highlighted in yellow background color.

We hope the reviewers will agree that our revisions have substantially improved and clarified our manuscript.

Sincerely,

On behalf of all other authors.

Reviewer #1:

Major issues

1. The introduction frequently discusses performance monitoring broadly, while the empirical analyses focus specifically on error monitoring (ERN). It would strengthen the paper to narrow the framing or explicitly note that only one aspect of performance monitoring is addressed.

Response: We thank the reviewer for this constructive comment. In response, we have revised the Introduction to specify that error monitoring is one component of performance monitoring and that our empirical analyses focus explicitly on error monitoring as indexed by the ERN. We believe this clarification improves the overall clarity of the manuscript.

2. Hypotheses center on EMA indices of desire, conflict, and enactment, yet the introduction provides relatively little theoretical or empirical context for these measures. More discussion of prior EMA research on self-control would give the hypotheses a stronger foundation.

Response: We agree and have revised the Introduction to include a more detailed discussion of prior EMA research on self-control, which reads as follows:

“Self-control refers to the ability to regulate behavior, thoughts and emotions in accordance with a specific goal or personal standard by changing or overriding competing response tendencies, desires or temptations [3,4]. Self-control includes both inhibitory and initiatory components and is crucial for adaptive behavior and successful goal pursuit, especially in contexts influenced by affective states [5-8]. A broader concept closely related to self-control is self-regulation, which includes goal setting, monitoring for the need of control, and implementing control processes according to set goals [9,10]. State self-control is often parsed and assessed using ecological momentary assessment of desires, associated goal conflict and behavioral outcomes [4,11,12], showing that higher desire strength was predictive of behavior enactment, whereas higher conflict strength was predictive of the behavior not being enacted.”

Literature

- Goschke, T. & Job, V. The willpower paradox: possible and impossible conceptions of self-control. *Perspectives on Psychological Science* **18**, 1339-1367, doi:10.1177/1745691622114 (2023).
- Hofmann, W., Baumeister, R. F., Forster, G. & Vohs, K. D. Everyday temptations: an experience sampling study of desire, conflict, and self-control. *J Pers Soc Psychol* **102**, 1318-1335, doi:10.1037/a0026545 (2012).
- de Ridder, D. T., de Boer, B. J., Lugtig, P., Bakker, A. B. & van Hooft, E. A. Not doing bad things is not equivalent to doing the right thing: Distinguishing between inhibitory and initiatory self-control. *Personality and Individual Differences* **50**, 1006-1011, doi:10.1016/j.paid.2011.01.015 (2011).
- Inzlicht, M., Werner, K. M., Briskin, J. L. & Roberts, B. W. Integrating models of self-regulation. *Annual review of psychology* **72**, 319-345, doi:10.1146/annurev-psych-061020-105721 (2021).
- de Ridder, D. T., Lensvelt-Mulders, G., Finkenauer, C., Stok, F. M. & Baumeister, R. F. Taking stock of self-control: A meta-analysis of how trait self-control relates to a wide range of behaviors. *Personality and Social Psychology Review* **16**, 76-99, doi:10.1177/1088868311418749 (2012).
- Inzlicht, M., Bartholow, B. D. & Hirsh, J. B. J. T. i. c. s. Emotional foundations of cognitive control. **19**, 126-132 (2015).
- Baumeister, R. F. & Heatherton, T. F. Self-regulation failure: An overview. *Psychological inquiry* **7**, 1-15, doi:10.1207/s15327965pli0701_1 (1996).
- Carver, C. & Scheier, M. *On the self-regulation of behavior*. (Cambridge, 1998).
- Schrader, S. W. & Tangney, J. P. Rethinking the Measurement of Self-Control: Distinguishing Among Self-Control Capacity, Urge Intensity, and Behavior Outcomes. *Basic and Applied Social Psychology* **47**, 37-46, doi:10.1080/01973533.2024.2415920 (2025).
- Hofmann, W., Vohs, K. D. & Baumeister, R. F. What people desire, feel conflicted about, and try to resist in everyday life. *Psychological science* **23**, 582-588, doi:10.1177/0956797612437426 (2012).

3. The power analysis states that three predictive effects were the focus, but it is not clear which effects these were or on what empirical basis the assumed effect size ($f^2 = 0.05$) was chosen. Since no identical studies exist in the literature, a clearer justification is needed.

Response: We thank the reviewer for this helpful comment.

For the power analysis, we focused on the three predictors central to our hypotheses: impulsivity, compulsivity, and ERN. Impulsivity and compulsivity are theorized to influence self-control directly and to moderate the association between ERN and self-control, which is tested via the three-way interaction ($ERN \times impulsivity \times compulsivity$) in the primary models. The assumed effect size was set at $f^2 = 0.05$, slightly smaller than that reported in previous research on the connection between error-related brain activity and self-control (Krönke et al., 2018; $\theta = 0.25$, corresponding to $f^2 \approx 0.07$), to ensure adequate statistical power. We have added this information in the corresponding section of the manuscript to clarify the rationale underlying the power analysis (see page 7).

Literature

- Krönke, K.-M. et al. Monitor yourself! Deficient error-related brain activity predicts real-life self-control failures. *Cognitive, Affective, & Behavioral Neuroscience* **18**, 622-637, doi:10.3758/s13415-018-0593-5 (2018).

4. The flanker task provided both speed- and accuracy-related feedback. Prior work indicates that external correctness feedback can reduce the salience of internal error-monitoring signals. This potential issue should be discussed as a limitation when interpreting ERN results.

Response: We thank the reviewer for raising this important point regarding potential interactions between external feedback and internal error monitoring signals. We agree that the presence of external feedback in flanker tasks can, in principle, influence ERN amplitude. In our study, however, the task design ensures that feedback is only informative under specific conditions: incorrect responses are always followed by negative feedback that is congruent with internal error-monitoring, whereas for correct responses, negative feedback is provided only for the slowest 20%

of trials. Thus, for incorrect responses, the initial motor action remains the most informative performance signal, in line with the “first indicator hypothesis” (Stahl, 2010). In our data, hand errors (wrong button presses) elicited a clear increase in ERN amplitude, but not in FRN amplitude, while timing errors showed distinct patterns depending on reaction. As the ERN in the current analyses was exclusively based on erroneous responses, it should reflect the processing of the internal response, rather than being overridden or diminished by subsequent feedback. We now include a brief discussion of this consideration as a potential limitation, while emphasizing that our design and the observed ERP patterns make it unlikely that ERN measurements were substantially affected by feedback: “Although our flanker task included external feedback, its potential interaction with internal error monitoring is likely minimal. Since incorrect responses were always followed by negative feedback, rendering it relatively uninformative, the robust ERN observed in our data likely reflects internal error processing, in line with the first indicator hypothesis (Stahl, 2010).”

Literature

Stahl, J. Error detection and the use of internal and external error indicators: An investigation of the first-indicator hypothesis. *International journal of psychophysiology* **77**, 43-52, doi:10.1016/j.ijpsycho.2010.04.005 (2010).

5. The three-way ERN × impulsivity × compulsivity interaction is statistically significant but likely small in effect size. More detail on robustness checks (e.g., bootstrapped confidence intervals, sensitivity analyses) would increase confidence that this effect is not fragile.

Response: We thank the reviewer for this question. To address concerns regarding robustness, we conducted additional parametric bootstrapping analyses. Specifically, we generated 2,000 parametric bootstrap samples of the fitted generalized linear mixed-effects model and computed percentile-based 95% confidence intervals for all fixed effects. The bootstrap confidence interval for the ERN × impulsivity × compulsivity interaction closely matched the Wald-based estimate and did not include zero ($\beta = -0.20$, 95% bootstrap CI $[-0.38, -0.01]$), indicating a stable sampling distribution of the effect despite its modest size.

In addition, the direction and magnitude of the interaction were consistent across estimation approaches and robust to subset-perturbation analyses (30 iterations, with 20 participants randomly removed), suggesting that the observed effect reflects a reliable, albeit small, moderating pattern rather than a statistical artifact.

We added these details to the Results section and provide the full bootstrap, optimizer and subset-perturbation analyses results in the Supplementary Materials.

6. I have the impression that the manuscript doesn't explicitly say how ERN was entered into the models. Did the authors take each participant's mean ERN amplitude (e.g., averaged across all errors) and use this as the predictor? Or did they, for example, split by gain/loss context and then average? Did they consider trial-level ERN values at all?

Response: ERN amplitudes were computed at the participant level by averaging across all errors in incongruent trials and entered into the models as participant-level predictors. For context-specific analyses, ERN amplitudes were averaged separately for gain and loss contexts. Trial-level ERN values were not modeled. We have clarified this procedure in the first paragraph of the Data analysis section (see page 9).

Minor issues

1. The EEG was re-referenced to the average of all electrodes. While this approach is common, mastoid (or near-mastoid, e.g., TP9 & TP10) references are often preferred for midline components such as the ERN and CRN. The authors should justify their choice.

Response: We chose the average reference scheme because we used a high-density montage and aimed to minimize bias from any single reference electrode. Previous work (e.g., Sandre et al., 2020) indicates that average and mastoid references yield comparable, and in some cases higher, reliability, internal consistency and temporal stability for ERN and CRN measures. We have added a brief justification to the corresponding section of the manuscript (see page 9).

Literature

Sandre, A., Banica, I., Riesel, A., Flake, J., Klawohn, J., & Weinberg, A. Comparing the effects of different methodological decisions on the error-related negativity and its association with behaviour and gender. *International Journal of Psychophysiology* **156**, 18-39, doi:10.1016/j.ijpsycho.2020.06.016 (2020).

2. The ERN and CRN waveforms are canonical (Fig. S2), but providing an additional topographic map (for the ERN–CRN difference) would help confirm that the observed effects show the expected scalp distribution.

Response: We have added additional topographical maps to the new Figure 2 in the main manuscript. Specifically, Figure 2D now displays response-locked event-related potentials (ERPs) for both contexts, gain, and loss context, showing the time course of incongruent error and correct trials averaged across Fz, FCz, and Cz with shaded areas representing the between subject SEM. In addition, topographical maps now depict mean response-locked activity at the mean ERN latency (54 ± 10 ms), shown separately for error and error-correct trials. These additions confirm the canonical midline scalp distribution of the ERN.

3. Please report the mean, minimum, and maximum number of trials per condition (after preprocessing and artifact rejection).

Response: We now include a detailed overview of the number of trials per trial type (after preprocessing and artifact rejection) in Table S3. We additionally reference this overview in the Methods section of the manuscript.

4. The discussion devotes substantial space to speculating about why impulsivity did not predict self-control. While some speculation is appropriate, this section could be tightened and balanced by acknowledging the possibility of genuinely null effects.

Response: We thank the reviewer for this suggestion. We have revised the discussion to explicitly acknowledge the possibility of a genuine null effect regarding the relationship between impulsivity and self-control. Additionally, we have shortened the speculative portion to make the discussion more concise and balanced. This section now reads as follows:

“Conversely, the association between impulsivity and self-control did not reach statistical significance in the present analyses. Although, a true null effect cannot be ruled out, it contrasts with common assumptions about the role of impulsivity in self-control [4]; the lack of a statistically significant association may reflect altered awareness of desires and conflicts in impulsive individuals. Previous studies have linked high impulsivity with reduced ERN, suggesting impaired

error and conflict monitoring [73]. Additionally, differences in subjective goals and personal standards or the valuation of what behavior conflicts with these may lead impulsive individuals to perceive fewer conflicts between desires and long-term goals [10], or even view enacted temptations positively, resulting in fewer reported failures of self-control [42]. Thus, trait-related deficits may be obscured when self-control is assessed via self-report, particularly if individuals differ in how they define or evaluate self-control based on their personal goals or standards.”

Literature

- Hofmann, W., Baumeister, R. F., Forster, G. & Vohs, K. D. Everyday temptations: an experience sampling study of desire, conflict, and self-control. *J Pers Soc Psychol* **102**, 1318-1335, doi:10.1037/a0026545 (2012).
- Arora, P. & Varshney, S. Analysis of k-means and k-medoids algorithm for big data. *Procedia Computer Science* **78**, 507-512 (2016).
- Carver, C. & Scheier, M. *On the self-regulation of behavior*. (Cambridge, 1998).
- Jansen, M. & de Bruijn, E. Mistakes that matter: An event-related potential study on obsessive-compulsive symptoms and social performance monitoring in different responsibility contexts. *Cognitive, affective & behavioral neuroscience*, doi:10.3758/s13415-020-00796-3 (2020).

5. It would be useful to clarify how missing or skipped EMA prompts were handled in the analyses.
Response: We thank the Reviewer for this comment and added a short statement clarifying the handling of missing data in the Data analysis section: “Missing or skipped EMA prompts were handled via listwise deletion at the level of individual observations. Robustness checks (see Supplement 3) were conducted for the multilevel models and indicated stable parameter estimates.”

6. Since the cluster analysis is exploratory, it should be framed more as hypothesis-generating than as evidence of distinct subgroups.

Response: We agree with the reviewer and have revised the framing of the cluster analysis in both the Results and Discussion sections to clearly emphasize its exploratory nature.

Reviewer #3:

While the authors give a good (if standard) description of self-control in their introduction I think it could be more nuanced given some recent claims. First, self-control isn't directly measured - impulsivity, compulsivity and lack of perseverance/premeditation are taken as indicators of self-control. But, as the originators of the self control scale most commonly used in the literature (Tangney et al) have recently suggested, these things might better reflect "failures" of self-control rather than any specific underlying mechanism. (Schrader, S. W., & Tangney, J. P. (2025). Rethinking the Measurement of Self-Control: Distinguishing Among Self-Control Capacity, Urge Intensity, and Behavior Outcomes. *Basic and Applied Social Psychology*, 47(1), 37-46.) I see throughout that failures of self-control are the focus, but some treatment of how this is (poorly) measured might be warranted. A tripartite distinction between goal initiation, goal persistence and inhibitory control may be a more nuanced way of exploring self-control (I think initiation is missing here unless you assume premeditation to be a measure of this). I don't think this is damning to the current paper - I just think a little more nuance is warranted.

Response: We thank the reviewer for this thoughtful and nuanced comment and appreciate the opportunity to clarify and strengthen the conceptual framing of our manuscript. We would like to clarify that the present study does not conceptualize impulsivity, compulsivity, or related symptom dimensions or traits as measures of self-control. Rather, these traits were included as moderators that provide affective-motivational context – given that these dimensions are likely rooted in affective-motivational dysregulation – for daily self-control processes assessed via ecological momentary assessment (EMA).

Our EMA approach captures urge intensity (i.e. desire strength), experienced goal-desire conflict, and behavioral outcomes, thereby focusing on interventive self-control as it unfolds in daily life rather than on self-control capacity per se. This process-oriented assessment of state-self-control (and its components) via EMA is explicitly recommended by Schrader and Tangney (2025). In line with this distinction, our measure of self-control focuses exclusively on interventive self-control – that is, the mobilization of control in the presence of temptation. We also explicitly acknowledge this is a limitation in the discussion already and note that future work should examine preventive self-control and strategic goal pursuit to capture the full spectrum of self-control processes. To make these distinctions even clearer for readers, we have revised the Introduction to further clarify our conceptualization of self-control and its relation to broader self-regulation frameworks. In addition, we have added a clarifying sentence to the Methods section stating: “Consistent with recent distinctions between self-control capacity, urge intensity, and behavioral outcomes (Schrader & Tangney, 2025), the present study does not aim to measure self-control capacity directly. Instead, we adopted a process-oriented approach.” We believe these revisions build directly on the reviewer’s suggestions and more clearly position our work within contemporary theoretical accounts of self-control.

Literature

Schrader, S. W. & Tangney, J. P. Rethinking the Measurement of Self-Control: Distinguishing Among Self-Control Capacity, Urge Intensity, and Behavior Outcomes. *Basic and Applied Social Psychology* 47, 37-46, doi:10.1080/01973533.2024.2415920 (2025).

Second, recent work from Eisenberg et al (2019) suggests that surveys and not tasks like the flanker, do a better (although not dramatically better) job of predicting real-world outcomes of self-regulatory control issues. (Eisenberg, I. W., Bissett, P. G., Zeynep Enkavi, A., Li, J., MacKinnon, D. P.,

Marsch, L. A., & Poldrack, R. A. (2019). Uncovering the structure of self-regulation through data-driven ontology discovery. *Nature communications*, 10(1), 2319.) I think this is relevant here too. Do we even know if ERNs (to tasks, which don't predict well) are predictive of real-world self-regulatory challenges (i.e., do ERNs predict drug and alcohol misuse as just one example?). Again, this is not damning to this paper, but I think it is worth including in the discussion at least as a potential limitation of this (and much of the self-control) work. The everyday reports of self-control challenges get at this somewhat but are not quite as direct as asking whether ERNs (vs. trait metrics collected) do a better (or even equivalent) job of predicting outcomes like substance abuse (which was prominently mentioned in the introduction).

Response: We thank the reviewer for this comment and appreciate the suggestion to explicitly consider the limitations of ERN-based measures in predicting real-world self-regulatory outcomes. We are aware that task-based measures typically explain only a small proportion of variance in everyday behavior, consistent with the findings from Eisenberg et al. (2019) suggesting that surveys often outperform laboratory tasks in predicting real-world self-regulatory challenges.

In response to this comment, we have revised the Introduction to highlight evidence that higher ERN amplitudes have been associated with better stress regulation, which is even reflected in lower cortisol (Compton et al., 2008; Compton et al., 2011; Compton et al., 2013), and self-regulation skills (Checa et al., 2014), and to note that the functional relevance of the ERN may be moderated by trait dimensions rooted in affective-motivational dysregulation (Bartholow, 2026). We also included findings showing that larger ERN amplitudes predict less risky alcohol use in adolescents (Boer et al. 2025) and successful abstinence after treatment in cocaine use disorder (Forster et al., 2026).

We acknowledge, as suggested, that the predictive power of ERN for daily-life behaviors, including substance use, is limited, and we now more explicitly note this in the discussion and Limitations section. Interestingly, recent evidence suggests that integrating EEG and fMRI measures of error processing may substantially increase its predictive power for future anxiety, even outperforming demographic variables and baseline anxiety (Valadez et al., 2025). At the same time, our study focuses on understanding the underlying processes of self-control and adaptive behavior by examining error processing and motivational salience in a fine-grained, process-oriented way. This approach provides insight into which stages of adaptive behavior— such as performance monitoring or conflict detection, and control implementation – may fail, thereby informing interventions aimed at improving self-regulation in both healthy and clinical populations.

Literature

- Compton, R. J. et al. Error-monitoring ability predicts daily stress regulation. *Psychological Science* **19**, 702-708, doi:10.1111/j.1467-9280.2008.02145.x (2008).
- Compton, R. J. et al. Neural and behavioral measures of error-related cognitive control predict daily coping with stress. *Emotion* **11**, 379, doi:10.1037/a0021776 (2011).
- Compton, R. J., Hofheimer, J. & Kazinka, R. Stress regulation and cognitive control: evidence relating cortisol reactivity and neural responses to errors. *Cognitive, Affective, & Behavioral Neuroscience* **13**, 152-163, doi:10.3758/s13415-012-0126-6 (2013).
- Checa, P., Castellanos, M., Abundis-Gutiérrez, A. & Rosario Rueda, M. Development of neural mechanisms of conflict and error processing during childhood: implications for self-regulation. *Frontiers in psychology* **5**, 326, doi:10.3389/fpsyg.2014.00326 (2014).
- Bartholow, B. D. Motivational Significance of Control Failures as a Window on Risk for Problematic Alcohol Involvement. *Biological psychiatry global open science* **6**, 100658, doi:10.1016/j.bpsgos.2025.100658 (2026).

- Boer, O. D., El Marroun, H., Ultanir, D. & Franken, I. H. Adolescent risky alcohol use is associated with electrophysiological markers of error processing: Findings from a large cohort study. *Biological Psychiatry Global Open Science*, 100615, doi:10.1016/j.bpsgos.2025.100615 (2025).
- Forster, S. E., Forman, S. D., Dickey, M. W., Siegle, G. J. & Steinhauer, S. R. Baseline Electrophysiological Markers of Reward and Error Processing are Associated with Improved Outcomes in Prize-Based Contingency Management. *Drug and Alcohol Dependence*, 113003, doi:10.1016/j.drugalcdep.2025.113003 (2025).
- Valadez, E. A. *et al.* Integrating multimodal neuroimaging of error monitoring to estimate future anxiety in adolescents. *JAMA Network Open* 8, e2539133-e2539133, doi:10.1001/jamanetworkopen.2025.39133 (2025).

With respect to the cluster analysis, did cluster 1 report fewer "failures" or just fewer situations demanding self-control overall? I know the measure used is a composite, but composites, like difference scores, hide somewhat the contribution of things like base rate of actual conflicts experienced.

Response: We have clarified the operationalization of the outcome measure in the data analysis section of the manuscript:

"The outcome was operationalized as enactments of conflict-laden desires divided by the number of questionnaires participants had responded to". Thus, the measure reflects the relative frequency of enactments rather than the absolute number of failures or the base rate of conflicts. Thus, the differences between clusters are not attributable merely to fewer situations demanding self-control, but to differences in enactments relative to reporting opportunities.

Also, I was expecting some exploration of how these distinct clusters might then relate to the ERN amplitude? Seems a glaring oversight to not even discuss this.

Response: We thank the reviewer for this question.

In response, we conducted additional analyses to exploratorily examine whether cluster membership moderates the association between ERN amplitude and self-control failures in daily life. Importantly, these analyses were explicitly framed as hypothesis-generating and were designed to complement the pre-registered dimensional analyses reported earlier in the manuscript. Consistent with the original moderation analyses showing that higher impulsivity and compulsivity attenuate the association between the ERN and self-control, the cluster-based analyses revealed a similar pattern. Specifically, larger ERN amplitudes were associated with fewer self-control failures in the cluster characterized by low compulsivity, anxiety and worrying, whereas the association was attenuated in the cluster characterized by high expressions of these symptoms or traits. Extending this exploratory approach further indicated that impulsivity additionally weakened the ERN-self-control association within the high symptom cluster, while no such effect was observed in the low symptom cluster.

These results are now reported in the Results section and discussed in relation to the original dimensional findings, thereby demonstrating conceptual convergence between the exploratory cluster-based analyses and the pre-registered trait-based models. Detailed results are provided in Supplement 6.

Minor points:

line 54: SUD (I presume substance use disorder?) is not spelled out - probably don't need to spell out OCD, but is SUD really as highly used and known? In general, acronyms help the writer far

more than the reader, so I would avoid them wherever possible. Turns out, you spell it out later in the Introduction - just move that forward to first usage (or eliminate the acronym altogether!).

Response: We have revised the manuscript accordingly by spelling out substance use disorder at its first occurrence and ensuring consistency throughout the Introduction.

The introduction in general puts forward a lot of causal models (e.g., higher ERN=higher response to errors=leads to increased demand for self-control="thereby producing more effective self-control" as just one example). Do we really have the data to speak to such causal arrows? I might temper this language to be about "relations" as opposed to "leads to" type language. In particular, at many time points it reads like the ERN causes self-control challenges - I'm not sure that's what the authors really want to suggest is it?

Response: We have carefully reviewed the Introduction and revised the language to reflect associative and theorized correlational rather than causal associations. Specifically, we now describe the framework in terms of performance monitoring - recruitment of control – self-control, emphasizing relations and theorized links without implying direct causation. This adjustment clarifies the intended theoretical perspective while avoiding overstatement of causal inference.

line 188: I don't think an effect of $p=0.258$ should be talked about as a "non-significant positive effect" - it's just plain ol' non-significant at that point, no?

Response: We have updated the text accordingly and now refer to the result simply as "non-significant".

Not sure how viable this is (some LLM analysis perhaps given the size of the data set) - but I was curious as to the nature of the conflicting desires that participants reported experiencing in the 7-day period. In part, this is to determine the "seriousness" of the conflicts - if they were relatively minor (as I suspect they would be in this sample) then we might expect more dramatic effects in pathological populations like obsessive-compulsive or substance use disorders. Be worth briefly commenting on.

Response: We have added a section to the Supplement (Supplement 5) addressing desire types and associated conflict. Overall, these data suggest that while participants experienced conflicts, the majority were related to common daily activities rather than extreme or pathological behaviors. As such, the conflicts appear relatively mild in this non-clinical sample, supporting the expectation that more pronounced or serious conflicts might be observed in clinical populations such as individuals with obsessive-compulsive disorder or substance use disorders.

"When considering conflict specifically, the desire types with the highest number of reported conflicts were similar routine behaviors: eating ($n = 697$ conflicts), resting ($n = 475$), sleeping ($n = 418$), using the internet ($n = 306$), and watching TV or streaming ($n = 213$). Average conflict strength was highest for desires that may involve self-control or social regulation rather than physiological needs: misbehaving (mean = 4.53, SD = 1.36), gambling (mean = 4.45, SD = 1.37), socializing (mean = 4.22, SD = 1.51), resting (mean = 4.18, SD = 1.40), and sleeping (mean = 4.15, SD = 1.40)."

Response Letter:

Dear Editors and Reviewers,

First of all, we sincerely thank the reviewers for their thoughtful comments and suggestions for revising our manuscript. We greatly appreciate the time and effort invested in providing detailed and constructive feedback.

In the following, we respond point by point to the remaining issues raised. The reviewers' comments are reproduced below and followed by our responses (in italic font for clarity).

We hope that the remaining revisions have adequately addressed all remaining concerns and that the manuscript has been substantially improved and clarified as a result.

Sincerely,

On behalf of all co-authors.

Reviewer #1:

The authors have addressed my previous concerns carefully, and I appreciate the additional robustness analyses and methodological clarifications. I have two remaining minor requests prior to acceptance, concerning (1) the strength of interpretation in the Abstract, and (2) the color scheme used for the ERN topographies.

(1) In particular, statements such as “These findings demonstrate that impulsivity and compulsivity jointly reduce the behavioral relevance of performance monitoring” and “underscoring the importance of accounting for trait interactions into cognitive and neural models of self-control” (Abstract) appear somewhat overconfident given the correlational design and the modest size of the three-way interaction. I recommend replacing such phrasing with more cautious language and explicitly acknowledging that mechanistic interpretations remain speculative.

Response: We thank the reviewer for this constructive comment. In response, we have revised the wording in the Abstract to adopt a more cautious tone. The revised version now explicitly acknowledges the correlational nature of the design and clarifies that mechanistic interpretations remain speculative. We believe this change appropriately tempers the strength of the conclusions while preserving the core contribution of the findings.

(2) In Fig. 1C and Fig. 2D, the ERN topographies are displayed using a color scheme in which negative values are shown in red and positive values in blue. This is opposite to common ERP conventions, where negative amplitudes are typically depicted in blue and positive in red, and may therefore be misleading, especially given that the ERN is a negative-going component. I recommend either adopting a conventional color mapping or, at minimum, making the color scale more prominent (e.g., larger values / clearer labeling) to ensure that the polarity assignment is immediately apparent to the reader.

Response: We appreciate this helpful suggestion. We have revised the figures so that negative amplitudes are now displayed in blue and positive amplitudes in red.

Reviewer #3:

The authors have done a good job addressing my initial concerns. I think the piece is ready for publication.

Response: We thank the reviewer for the positive evaluation of our manuscript and for the constructive feedback provided during the review process.